# Assessor360: Multi-sequence Network for Blind Omnidirectional Image Quality Assessment

**Tianhe Wu**[1*]**, Shuwei Shi**[1,2,*]**, Haoming Cai**[3]**, Mingdeng Cao**[2]**,**
**Jing Xiao**[4]**, Yinqiang Zheng**[2]**, Yujiu Yang**[1†]

[1] Shenzhen International Graduate School, Tsinghua University
[2] The University of Tokyo   [3] University of Maryland, College Park   [4] Pingan Group
{wth22, ssw20}@mails.tsinghua.edu.cn, cmd@g.ecc.u-tokyo.ac.jp
hmcai@umd.edu, xiaojing661@pingan.com.cn, yqzheng@ai.u-tokyo.ac.jp
yang.yujiu@sz.tsinghua.edu.cn

## Abstract

Blind Omnidirectional Image Quality Assessment (BOIQA) aims to objectively assess the human perceptual quality of omnidirectional images (ODIs) without relying on pristine-quality image information. It is becoming more significant with the increasing advancement of virtual reality (VR) technology. However, the quality assessment of ODIs is severely hampered by the fact that the existing BOIQA pipeline lacks the modeling of the observer's browsing process. To tackle this issue, we propose a novel multi-sequence network for BOIQA called Assessor360, which is derived from the realistic multi-assessor ODI quality assessment procedure. Specifically, we propose a generalized Recursive Probability Sampling (RPS) method for the BOIQA task, combining content and details information to generate multiple pseudo viewport sequences from a given starting point. Additionally, we design a Multi-scale Feature Aggregation (MFA) module with a Distortion-aware Block (DAB) to fuse distorted and semantic features of each viewport. We also devise Temporal Modeling Module (TMM) to learn the viewport transition in the temporal domain. Extensive experimental results demonstrate that Assessor360 outperforms state-of-the-art methods on multiple OIQA datasets. The code and models are available at https://github.com/TianheWu/Assessor360.

## 1   Introduction

With the development of VR-related techniques, viewers can enjoy a realistic and immersive experience with head-mounted displays (HMDs) [3, 20] by perceiving 360-degree omnidirectional information. However, the acquired omnidirectional image, also named panorama, is not always of high quality [12, 60]. Degradation may be introduced in any image processing [44, 58, 42], leading to low-quality content that may be visually unpleasant for users. Consequently, developing suitable quality metrics for panoramas holds considerable importance, as they can be utilized to guide research in omnidirectional image processing and maintain high-quality visual content.

Generally, most ODIs are stored in the equirectangular projection (ERP) format, which exhibits considerable geometric deformation at different latitudes. This distortion can have a negative impact on quality assessment. Therefore, as shown in Figure 1 (a), many researchers [36, 17, 43, 58, 14, 60] explore viewport-based methods by projecting the original ERP image into many undeformed viewports (actual content observed by users) and aggregate their features with 2D IQA method as the ODI quality score. However, this conventional viewport-based pipeline lacks the modeling of the

---

[*]Tianhe Wu and Shuwei Shi contribute equally to this work.
[†]Corresponding author.

37th Conference on Neural Information Processing Systems (NeurIPS 2023).

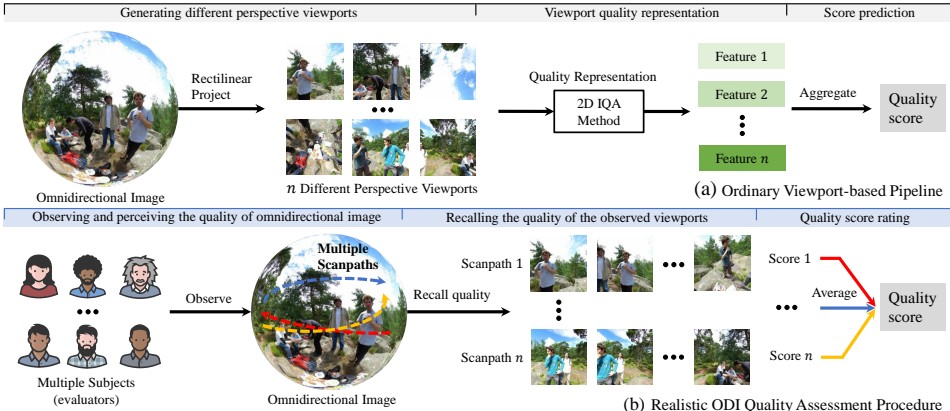

Figure 1: Illustration of the existing ordinary viewport-based pipeline (a) and the realistic omnidirectional image quality assessment procedure (b). Plenty of existing viewport-based methods follow pipeline (a) which is inconsistent with authentic assessment procedure (b), causing the predicted quality score to be far from the human perceptual quality score.

observer's browsing process, causing the predicted quality to be far from the human perceptual quality, especially when the ODI containing non-uniform distortion (such as stitching) [33, 12, 36, 61, 37]. In fact, during the process of observing ODIs, viewports are sequentially presented to viewers based on their browsing paths, forming a viewport sequence.

While the viewport sampling techniques proposed by Sui *et al.*[33] and Zhang *et al.*[53] can generate viewports with a fixed sequential order for ODIs, their methods lack the ability to generate versatile sequences. Specifically, their methods will produce identical sequences for a given starting point, and the sequential order of viewports will remain constant across different ODIs. This behavior is inconsistent with the observations of multiple evaluators in realistic scenarios, leading to an inability to provide subjectively consistent evaluation results. To address this issue, a logical approach is to employ the scanpath prediction model for generating pseudo viewport sequences on the ODI without an authentic scanpath. However, current scanpath prediction methods [47, 24, 32, 1, 34] are developed for undistorted ODIs and mainly focus on high-level regions. As evaluators are required to provide an accurate quality score for an ODI, their scanpaths are distributed across both low-quality and high-detail regions. This makes it potentially unsuitable for direct application in BOIQA task where all ODIs are distorted.

To move beyond these limitations, inspired by the realistic multi-assessor ODI quality assessment procedure, shown in Figure 1 (b), we first propose a multi-sequence network called Assessor360 for BOIQA, which can simulate the authentic data scoring process to generate multiple viewport sequences (corresponding to multiple scanpaths). Specifically, we propose Recursive Probability Sampling (RPS) to generate multiple pseudo viewport sequences, combining semantic scene and local distortion characteristics. In particular, based on Equator-guided Sampled Probability (ESP) and Details-guided Sampled Probability (DSP), RPS will generate different viewport sequences for the same starting point (details in Section 3.2). Furthermore, we develop Multi-scale Feature Aggregation (MFA) with Distortion-aware Block (DAB) to effectively fuse viewport semantic and distorted features for accurate quality perception. The Temporal Modeling Module (TMM) is devised by applying GRU [5] module and MLP layers to learn the viewports temporal transition information in a sequence and regress the aggregated features to the final score. Extensive experiments demonstrate the superiority and effectiveness of proposed Assessor360 on multiple OIQA datasets (MVAQD [18], OIQA [11], CVIQD [35], IQA-ODI [46], JUFE [12], JXUFE [33]). We summarize our contributions into four points:

- We propose Assessor360 for BOIQA, which can leverage multiple different viewport sequences to assess the ODI quality. To our knowledge, Assessor360 is the first pipeline to simulate the authentic data scoring process in ODI quality assessment.

- We propose an unlearnable method, Recursive Probability Sampling (RPS) that can combine semantic scene and local distortion characteristics to generate different viewport sequences for a given starting point.

- We design Multi-scale Feature Aggregation (MFA) and Distortion-aware Block (DAB) to characterize the integrated features of viewports and devise a Temporal Modeling Module (TMM) to model the temporal correlation between viewports.

- We apply our Assessor360 to two types of OIQA task datasets: one with real observed scanpath data and the other without. Extensive experiments show that our Assessor360 largely outperforms state-of-the-art methods.

## 2 Related Work

**Omnidirectional Image Quality Assessment.** Similarly to traditional 2D IQA, according to the reference information availability, OIQA can also be divided into three categories: full-reference (FR), reduced-reference (RR), and no-reference (NR) OIQA, also known as blind OIQA [2]. Due to the structural characteristics of the panorama and the complicated assessment process, OIQA has not matured as much as 2D-IQA [48, 22, 13, 19, 15, 29]. Some researchers extend the 2D image quality assessment metrics to the panorama, such as S-PSNR [51], CPP-PSNR [52], and WS-PSNR [37]. However, these methods are not consistent with the Human Visual System (HVS) and they are poorly consistent with perceived quality [60, 43]. Although WS-SSIM [61] and S-SSIM [4] consider some impacts of HVS, the availability of non-distortion reference ODIs severely hinders their applications in authentic scenarios.

Therefore, some deep learning-based BOIQA methods [60, 43, 36, 16, 53, 46] are devised to achieve better capabilities. Due to the geometric deformation present in ODIs in ERP format, many existing BOIQA methods [59, 46, 60, 17] follow a similar pipeline: sampling viewports in a particular way and simply regressing their features to the quality score. MC360IQA [36] first maps the sphere into a cubemap, then employs CNN to aggregate each cubemap plane feature and regress them to a score. ST360IQ [16] samples tangent viewports from the salient parts and uses ViT [10] to estimate the quality of each viewport. VGCN [43] migrates the graph convolution network and pre-trained DBCNN [56] to establish connections between different viewports. However, these methods ignore the vital effect of the viewport sequences generated in multiple observers' browsing process, which have been demonstrated by Sui *et al.*[33] and Fang *et al.*[12] on the perception of the ODI quality.

**Viewport Sampling Strategies in OIQA.** Existing viewport sampling strategies can be categorized into three modes: 1) Uniformly sampling without sequential order. Zhou *et al.*[59] and Fang *et al.*[12] uniformly extract viewports over the sphere. Jiang *et al.*[17] and Sun *et al.*[36] use cube map projection (CMP) and rotate the longitude to obtain several viewport groups. This sampling pattern will cover the full areas, whether they are significant or non-significant. 2) Crucial region sampling without sequential order. Xu *et al.*[43] leverage 2D Gaussian Filter [45] to acquire a heatmap and generate viewports with corresponding locations. Tofighi *et al.*[16] apply ATSal [6] to predict salient regions of the panorama, which gives help to sampling viewports. This mode incorporates HVS, where most of the sampled viewport might be observed in the browsing process. Nevertheless, both of the above methods only focus on the image content and do not concern the effect of the sequential order between viewports in the quality assessment process of panoramas. 3) Sampling viewports along the fixed direction. Sui *et al.*[33] first considers the effect of sequential order between viewports in the browsing process. They default observers rotate their perspective in a specific direction along the equator. Zhang *et al.*[53] further introduce ORB detection to capture key viewports and follow Sui *et al.*[33] default sampling direction to extract viewports. In fact, different evaluators will produce various scanpaths when viewing a panorama. Even if the same evaluator observes the same ODI twice, the scanpath will be different. However, their sampling methods cause the sequential order of the viewport to be fixed, and even for the different image contents, the sequential order of sampled viewports is the same. This creates a significant gap with people's actual browsing process, leading to unreasonable modeling of the viewport sequence.

## 3 Method

### 3.1 Overall Framework

As shown in Figure 2 (a), the proposed Assessor360 framework consists of Recursive Probability Sampling (RPS) scheme, Multi-scale Feature Aggregation (MFA) strategy, and Temporal Modeling Module (TMM). Given a degraded ODI $\mathcal{I}$, to be consistent with the authentic multi-assessor assessment, we initialize $N$ starting points $\mathcal{X} = \{(u_i, v_i)\}_{i=1}^N$, where $u_i \in [-90°, 90°]$ and $v_i \in [-180°, 180°]$ are the corresponding latitude and longitude coordinates. Then, we apply the proposed RPS strategy $\mathcal{G}$ with parameters $\Theta_g$ to generate multiple viewport sequences $\mathcal{S} = \{\{\mathcal{V}_j^i\}_{j=1}^M\}_{i=1}^N$, where $\mathcal{V}$ denotes

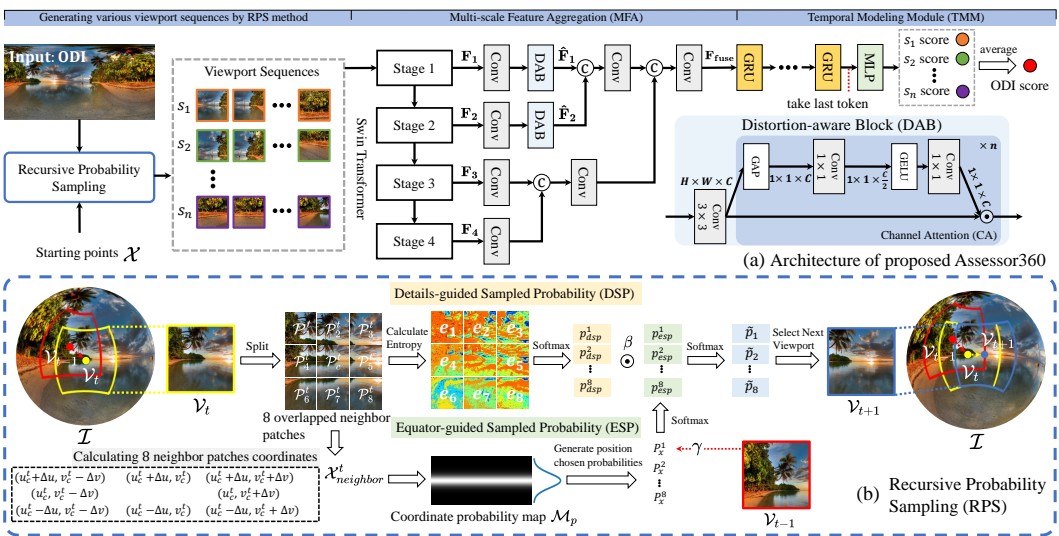

Figure 2: Architecture of proposed Assessor360 for BOIQA (a). Our Assessor360 consists of Recursive Probability Sampling (RPS) scheme (b), Multi-scale Feature Aggregation (MFA) strategy, and Temporal Modeling Module (TMM).

each viewport, $M$ denotes the length of each sequence. Next, to perceive the semantic scene and distortion in each viewport, multi-scale features are extracted from multi-stage layers of our method for quality assessment. We use MFA $\mathcal{F}$ with parameters $\Theta_f$ to represent this process. The features are aggregated from $\mathbb{R}^{H \times W \times C}$ to $\mathbb{R}^{1 \times 1 \times D}$. In this case, $C$ is the viewport's original dimensions, $H$ and $W$ are the height and width of the viewport, and $D$ denotes the embedding dimension. Then, we use TMM $\mathcal{H}$ with parameters $\Theta_h$ to model each sequence viewport temporal transition information and predict the final viewport sequence quality score. Finally, we average all predicted scores of each sequence as the ODI's quality $\mathcal{Q}_{\mathcal{I}}$. Overall, the whole process can be described as follows:

$$\mathcal{Q}_{\mathcal{I}} = \frac{1}{N}\sum_{i=1}^{N}\mathcal{H}(\mathcal{F}(\mathcal{G}(\mathcal{I}, \mathcal{X}; \Theta_g); \Theta_f); \Theta_h) \tag{1}$$

### 3.2 Recursive Probability Sampling

Recursive Probability Sampling (RPS) strategy can adaptively generate the probability of scene transition direction based on prior knowledge of semantic context and degraded features in ODI. It mainly consists of Equator-guided Sampled Probability (ESP) and Details-guided Sampled Probability (DSP). The viewport sequence is generated by selecting a certain starting point and sampling viewports based on generated probabilities.

**Preprocessing for generating probabilities.** As illustrated in [47], transition direction and distance are two important factors for locating the next viewport position. We first follow the theory of Moore neighborhood [28] and define $K$ neighbor ($K = 8$) transition directions from the center coordinate of the viewport. Following [31, 33], the transition distance $(\Delta u, \Delta v)$ is set to $(24°, 24°)$, avoiding sampling overly overlapped viewports. In details, given a current position $x_c^t = (u_c^t, v_c^t)$, we first generate the corresponding viewport $\mathcal{V}_t$ from $\mathcal{I}$ by rectilinear projection [49] $\mathcal{R}$. Then, we uniformly split $\mathcal{V}_t$ into $K + 1$ overlapped patches, including $K$ neighbor patches $\mathcal{P}^t = \{\mathcal{P}_i^t\}_{i=1}^{K}$ and one central patch $\mathcal{P}_c^t$ with height $\frac{H}{2}$ and width $\frac{W}{2}$. As shown in Figure 2 (b), the $K$ direction sampled coordinates $\mathcal{X}_{neighbor}^t = \{x_i^t\}_{i=1}^{K}$ can be calculated by the central patch $\mathcal{P}_c^t$ coordinate $x_c^t$ and $(\Delta u, \Delta v)$. During the browsing process, assessors are not only drawn to the high-level scenario but also focus on low-level texture and details regions [35, 11] to give a reasonable quality score. Therefore, according to the generalized prior content information and pixel-level details metric, we present ESP and DSP for each direction sampled coordinate in $\mathcal{X}_{neighbor}^t$. Then, we choose the next sampled viewport $\mathcal{V}_{t+1}$ position $x_c^{t+1}$ based on them.

**Equator-guided Sampled Probability (ESP).** Since the equator entails more scene information [7, 31], we introduce the prior equator bias [9] $\mathcal{M}$ to constrain the sampled viewport near the equator. Concretely, the prior equator bias obeys a Gaussian distribution with a mean of $0$ and a standard deviation of $0.2$ in latitude. Regions near the equator have higher sampled weights and regions close

**Algorithm 1** Viewport Sequence Generation (RPS Algorithm)

---

**Input:** $N$ starting points $\{x_i\}_{i=1}^N$; an ODI $\mathcal{I}$; rectilinear projection $\mathcal{R}$; ESP calculation function $\mathcal{F}_{esp}(\mathcal{X})$; DSP calculation function $\mathcal{F}_{dsp}(\mathcal{P})$; selecting function $\Gamma(\mathcal{X}|\tilde{p})$

**Output:** A set of $N$ length $M$ viewport sequences $\{s_i = \{\mathcal{V}_t\}_{t=1}^M\}_{i=1}^N$;

1: **for** $i = 1 \rightarrow N$ **do**
2:     Initialize the current coordinate $x \leftarrow x_i$
3:     **for** $t = 1 \rightarrow M$ **do**
4:         Generate viewport by the current coordinate $\mathcal{V} \leftarrow \mathcal{R}(x, \mathcal{I})$
5:         Split $\mathcal{V}$ to obtain overlapped neighbor patches $\mathcal{P}$ and calculate sampled coordinate $\mathcal{X}$
6:         Calculate ESP and DSP $p_{esp} \leftarrow \mathcal{F}_{esp}(\mathcal{X}), p_{dsp} \leftarrow \mathcal{F}_{dsp}(\mathcal{P})$
7:         Generate next viewport coordinate $x \leftarrow \Gamma(\mathcal{X}|\mathrm{Aggregate}(p_{esp}, p_{dsp}))$
8:     Sequentially gather generated $M$ viewports $\{\mathcal{V}_t\}_{t=1}^M$ as a viewport sequence $s_i$
9: Gather generated $N$ viewport sequences $\{s_i\}_{i=1}^N$ as the output

---

to the poles have relatively low sampled weights. We apply the softmax function to $\mathcal{M}$ to obtain the coordinate probability map $\mathcal{M}_p$ where each coordinate corresponds to a sampled probability. Then we take $\mathcal{X}_{neighbor}^t$ sampled probabilities $\{p_x^i\}_{i=1}^K$ from $\mathcal{M}_p$. Meanwhile, due to the inhibition of return (IOR) [27] where regions that have been fixated by the eyes have a lower probability of being fixated again in the near future, we multiply $p_x^k$ with a decreasing factor $\gamma$ where $k$ is the index of the viewport coordinate has been generated. Finally, we apply softmax function to calculate $p_{esp} = \{p_{esp}^i\}_{i=1}^K$. Overall, the ESP calculation function $\mathcal{F}_{esp}(\mathcal{X})$ can be defined as:

$$\mathcal{F}_{esp}(\mathcal{X}) = \mathrm{Softmax}(Z \cdot \mathcal{M}_p(\mathcal{X}_{neighbor}^t)), Z = \{1, \ldots, \gamma, \ldots, 1\}^K \tag{2}$$

**Details-guided Sampled Probability (DSP).**    To efficiently measure the texture complexity of each patch, we use pixel-level information entropy $\mathcal{E}$. The mathematical expression is:

$$\mathcal{E}(\mathcal{P}_i^t) = \sum_{m=1}^{H'} \sum_{n=1}^{W'} -p(\mathcal{P}_i^t[m, n]) \log_2 p(\mathcal{P}_i^t[m, n]) \tag{3}$$

where $\mathcal{P}_i^t[m, n]$ is the gray value of each pixel at position $(m, n)$ in the patch $\mathcal{P}_i^t$, and $p(\cdot)$ is the probability of occurrence of each gray value. After calculating the entropy of each neighbor patch, we normalize it with the softmax function to obtain $p_{dsp} = \{p_{dsp}^i\}_{i=1}^K$ for $\mathcal{X}_{neighbor}^t$. The DSP calculation function $\mathcal{F}_{dsp}(\mathcal{P})$ can be formulated as:

$$\mathcal{F}_{dsp}(\mathcal{P}) = \mathrm{Softmax}(\mathcal{E}(\mathcal{P}^t)) \tag{4}$$

**Generating viewport sequences.**    Ultimately, we multiply the ESP and DSP with a scale factor $\beta$ to get the integrated probability. We use the softmax function to obtain the final probability distribution and select the next viewport position $x_c^{t+1}$ according to it, which can be written as:

$$x_c^{t+1} = \Gamma(\mathcal{X}^t|\mathrm{Softmax}(p_{dsp} \cdot p_{esp} \cdot \beta)) \tag{5}$$

where $\Gamma(\mathcal{X}|\tilde{p})$ is the selecting function based on the set of probability $\tilde{p}$. We can recursively generate viewport $\mathcal{V}_{t+1}$ with $x_c^{t+1}$ and keep performing the RPS strategy until the number of viewports in the sequence reaches the desired length. The whole generation algorithm $\mathcal{G}$ is shown in Algorithm 1.

### 3.3 Multi-scale Feature Aggregation

To represent the semantic information and distortion pattern of each viewport, which are assumed as two key factors of quality assessment, we aggregate multi-scale features of the viewport. We first extract features from each stage in pre-trained Swin Transformer [23, 30]. The outputs of the first two stages $\mathbf{F_1}, \mathbf{F_2}$ are more sensitive to the distortion pattern, while the outputs of the last two stages $\mathbf{F_3}, \mathbf{F_4}$ tend to capture the abstract features (details in Supplementary Materials). Before fusing multi-scale features, we first employ four $1 \times 1$ convolution layers to reduce the feature dimension of the output to $D$. Then, to further emphasize the local degradation, we devise and apply the Distortion-aware Block (DAB) which includes a $3 \times 3$ convolution layer following $n$ Channel Attention (CA) operations to the reduced shallow features $\hat{\mathbf{F}}_1, \hat{\mathbf{F}}_2$. This operation can help the model to better perceive the distortion pattern in the channel dimension, achieving distortion-aware capacity. Finally, we concatenate and integrate these features with Global Average Pooling (GAP) and multiple $1 \times 1$ convolution layers to get the quality-related representation. After that, each aggregated feature $\mathbf{F_{fuse}}$ will be sent to TMM for viewport sequence quality assessment.

Table 1: Quantitative comparison of the state-of-the-art methods and proposed Assessor360. The best are shown in **bold**, and the second best (except ours) are underlined. Two baselines w/ ERP and w/ CMP mean that we replace input viewport sequences generated by RPS with ERP and CMP.

| Type | Method | MVAQD | | OIQA | | IQA-ODI | | CVIQD | |
|---|---|---|---|---|---|---|---|---|---|
| | | SRCC | PLCC | SRCC | PLCC | SRCC | PLCC | SRCC | PLCC |
| FR-IQA methods | PSNR | 0.8150 | 0.7591 | 0.3929 | 0.3893 | 0.4018 | 0.4890 | 0.8015 | 0.8425 |
| | SSIM [39] | 0.8272 | 0.7202 | 0.3402 | 0.2307 | 0.5014 | 0.5686 | 0.6737 | 0.7273 |
| | MS-SSIM [40] | 0.8032 | 0.7136 | 0.5750 | 0.5084 | 0.7434 | 0.8389 | 0.9218 | 0.9272 |
| | WS-PSNR [37] | 0.8152 | 0.7638 | 0.3829 | 0.3678 | 0.3780 | 0.4708 | 0.8039 | 0.8410 |
| | WS-SSIM [61] | 0.8236 | 0.5328 | 0.6020 | 0.3537 | 0.5325 | 0.7098 | 0.8632 | 0.7672 |
| | VIF [54] | 0.8687 | 0.8436 | 0.4284 | 0.4158 | 0.7109 | 0.7696 | 0.9502 | 0.9370 |
| | DISTS [8] | 0.7911 | 0.7440 | 0.5740 | 0.5809 | 0.8513 | 0.8723 | 0.8771 | 0.8613 |
| | LPIPS [55] | 0.8048 | 0.7336 | 0.5844 | 0.4292 | 0.7355 | 0.7411 | 0.8236 | 0.8242 |
| NR-IQA methods | NIQE [26] | 0.6785 | 0.6880 | 0.8539 | 0.7850 | 0.6645 | 0.5637 | 0.9337 | 0.8392 |
| | BRISQUE [25] | 0.8408 | 0.8345 | 0.8213 | 0.8206 | 0.8171 | 0.8651 | 0.8269 | 0.8199 |
| | PaQ-2-PiQ [50] | 0.3251 | 0.3643 | 0.1667 | 0.2102 | 0.0201 | 0.0419 | 0.7376 | 0.6500 |
| | MANIQA [48] | 0.5531 | 0.5718 | 0.4555 | 0.4171 | 0.2642 | 0.2776 | 0.6013 | 0.6142 |
| | MUSIQ [19] | 0.5436 | 0.6117 | 0.3216 | 0.3087 | 0.0565 | 0.0983 | 0.3483 | 0.3678 |
| | CLIP-IQA [38] | 0.5862 | 0.4941 | 0.2330 | 0.2531 | 0.0927 | 0.1929 | 0.4884 | 0.4347 |
| | LIQE [57] | 0.6837 | 0.7539 | 0.7634 | 0.7419 | 0.8551 | 0.9020 | 0.8594 | 0.8086 |
| | SSP-BOIQA [58] | 0.7838 | 0.8406 | 0.8650 | 0.8600 | - | - | 0.8614 | 0.9077 |
| | MP-BOIQA [17] | 0.8420 | 0.8543 | 0.9066 | 0.9206 | - | - | 0.9235 | 0.9390 |
| | MC360IQA [36] | 0.6605 | 0.6977 | 0.9071 | 0.8925 | 0.8248 | 0.8629 | 0.8271 | 0.8240 |
| | SAP-net [46] | - | - | - | - | 0.9036 | 0.9258 | - | - |
| | VGCN [43] | 0.8422 | 0.9112 | 0.9515 | 0.9584 | 0.8117 | 0.8823 | 0.9639 | 0.9651 |
| | AHGCN [14] | - | - | 0.9647 | 0.9682 | - | - | 0.9617 | 0.9658 |
| | baseline w/ ERP | 0.9076 | 0.9240 | 0.8961 | 0.8857 | 0.9098 | 0.9196 | 0.9330 | 0.9485 |
| | baseline w/ CMP | 0.8966 | 0.9324 | 0.9216 | 0.9170 | 0.9105 | 0.9122 | 0.9390 | 0.9412 |
| | **Assessor360** | **0.9607** | **0.9720** | **0.9802** | **0.9747** | **0.9573** | **0.9626** | **0.9644** | **0.9769** |

## 3.4 Temporal Modeling Module

The browsing process of ODI naturally leads to the temporal correlation. The recency effect indicates that users are more likely to evaluate the overall image quality affected by the viewports they have recently viewed, especially during prolonged exploration periods. To model this relation, we introduce the GRU module to learn the viewport transition. Due to the fact that the last token encodes the most recent information and the representation at the last time step involves the whole temporal relationships of a sequence, we use MLP layers to regress the last feature output by the GRU module to the sequence quality score.

## 4 Experiments

### 4.1 Implementation Details

We set the field of view (FoV) to the $110°$ following [12, 33]. We use pre-trained Swin Transformer [23] (base version) as our feature extraction backbone. The input viewport size $H \times W$ is fixed to $224 \times 224$. The number of viewport sequences $N$ is set to 3 and the length of each sequence $M$ is set to 5. We set the coordinates of $N$ starting points to be $(0°, 0°)$. The reduced dimension $D$ is 128 and the number of GRU modules is set to 6. The number of CA operations $n$ is 4. We set $\gamma = 0.7$ and $\beta = 100$ as decreasing factor and scale factor values respectively.

For a fair comparison, we randomly split $80\%$ ODIs of each dataset for training, and the remaining $20\%$ is used for testing following [53, 18, 12, 17]. To eliminate bias, we run a random train-test splitting process ten times and show the median result. We train 300 epochs with batch size 4 on CVIQD [35], OIQA [11], IQA-ODI [46], and MVAQD [18] datasets without the authentic scanpath data. Respectively, we compare our RPS with two advanced learning-based scanpath prediction methods ScanGAN360 [24] and ScanDMM [32] on JUFE [12] and JXUFE [33] datasets which have the authentic scanpath data. For optimization, we use Adam [21] and the learning rate is set to $1 \times 10^{-5}$ in the training phase. We employ MSE loss to train our model. We use Spearman rank-ordered correlation (SRCC) and Pearson linear correlation (PLCC) as the evaluation metrics.

### 4.2 Comparing with the State-of-the-art Methods

We conduct a comparative analysis of Assessor360 with eight FR methods and thirteen NR methods. The quantitative comparison results are presented in Table 1, demonstrating significant performance

Table 2: Cross-dataset validation SRCC and PLCC results of SOTA methods. These models (except WS-PSNR [37] and WS-SSIM [61]) are trained on CVIQD [35], OIQA [11] and MVAQD [18] datasets (80% set) and tested on three other datasets (full set).

| Method | CVIQD | | | OIQA | | | MVAQD | | |
|---|---|---|---|---|---|---|---|---|---|
| | OIQA | IQA-ODI | MVAQD | CVIQD | IQA-ODI | MVAQD | CVIQD | OIQA | IQA-ODI |
| | | | | SRCC | | | | | |
| WS-PSNR | 0.5027 | 0.4360 | 0.7225 | 0.7638 | 0.4360 | 0.7225 | 0.7638 | 0.5027 | 0.4360 |
| WS-SSIM | **0.5442** | 0.5032 | **0.7930** | 0.6625 | 0.5032 | **0.7930** | 0.6625 | 0.5442 | 0.5032 |
| MC360IQA | 0.4189 | 0.7114 | 0.0296 | 0.7044 | 0.5687 | 0.4081 | 0.0373 | 0.0025 | 0.0486 |
| VGCN | 0.2361 | 0.2875 | 0.2452 | 0.6932 | 0.3873 | 0.4682 | 0.4650 | 0.6227 | 0.3921 |
| **Assessor360** | 0.4597 | **0.8610** | 0.5640 | **0.8430** | **0.8751** | 0.6417 | **0.8756** | **0.7765** | **0.8646** |
| | | | | PLCC | | | | | |
| WS-PSNR | 0.4701 | 0.5468 | **0.6962** | 0.7895 | 0.5468 | **0.6962** | 0.7895 | 0.4701 | 0.5468 |
| WS-SSIM | 0.4363 | 0.5941 | 0.6246 | 0.6536 | 0.5941 | 0.6246 | 0.6536 | 0.4363 | 0.5941 |
| MC360IQA | 0.4295 | 0.7872 | 0.0404 | 0.7368 | 0.5930 | 0.4238 | 0.0430 | 0.0202 | 0.0646 |
| VGCN | 0.2582 | 0.3127 | 0.2467 | 0.5929 | 0.3551 | 0.2419 | 0.3420 | 0.4642 | 0.3870 |
| **Assessor360** | **0.5332** | **0.9032** | 0.5824 | **0.8636** | **0.9137** | 0.6565 | 0.7232 | **0.7287** | **0.8541** |

Table 3: Cross-dataset validation SRCC results of SOTA methods. These models are trained on MVAQD [18] (full set) and tested on CVIQD [35] and OIQA [11] all distortion data except MP-BOIQA (removing AVC on CVIQD).

| Testing set | MC360IQA | MP-BOIQA | **Assessor360** |
|---|---|---|---|
| OIQA | 0.2542 | 0.5043 | **0.6658** |
| CVIQD | 0.4749 | 0.7992 | **0.8994** |

Table 4: Quantitative comparison of using different viewport sequence generation methods on OIQA [11] and MVAQD [18].

| Generation Method | OIQA | | MVAQD | |
|---|---|---|---|---|
| | SRCC | PLCC | SRCC | PLCC |
| Random Generation | 0.9461 | 0.9444 | 0.9359 | 0.9543 |
| ScanGAN360 | 0.9705 | 0.9670 | 0.9493 | 0.9694 |
| ScanDMM | 0.9652 | 0.9634 | 0.9558 | 0.9612 |
| **RPS (Ours)** | **0.9802** | **0.9747** | **0.9607** | **0.9720** |

improvements across all datasets against the state-of-the-art methods. Note that these performance enhancements are obtained without training with the 2D IQA dataset, as employed in VGCN [43]. Specifically, Assessor360 outperforms VGCN by 3% in terms of SRCC on the OIQA dataset and shows a 1.2% increase in PLCC on the CVIQD dataset. Additionally, our method exhibits a notable improvement of up to 1.6% in SRCC compared to AHGCN [14].

## 4.3 Cross-Dataset Evaluation

To assess the generalization capability of our method, we perform cross-validation on the CVIQD [35] and OIQA [11] datasets and compare it against two widely used state-of-the-art methods: MC360IQA [36] and MP-BOIQA [17]. All models are trained on the MVAQD [18] dataset and subsequently tested on the CVIQD and OIQA datasets. The results, presented in Table 3, demonstrate that our method exhibits superior generalization performance compared to the other models. It is worthy to note that, there exists a significant domain gap between the CVIQD and MVAQD datasets. The CVIQD dataset primarily focuses on degradations such as JPEG, H.264/AVC, and H.265/HEVC, while the MVAQD dataset concentrates on JPEG, JP2K, HEVC, WN, and GB. This discrepancy in focus poses a challenge for MC360IQA and MP-BOIQA, as they use the same perspective viewports to evaluate different distorted ODIs, leading to poor generalization. In contrast, our proposed RPS overcomes this domain gap effectively by generating different viewport sequences for ODIs with different distortions. It consistently performs well, showcasing its effectiveness in addressing the challenges posed by diversely distorted ODIs.

Furthermore, to verify that the performance presented in Table 1 is not due to the overfitting, we retrain MC360IQA, VGCN [43] and Assessor360 on CVIQD, OIQA and MVAQD datasets, using 80% of the dataset for training and the remaining 20% for testing. We then select the model weights with the highest performance on the testset for cross-dataset testing for fair generalization comparisons. The results shown in the Table 2 clearly demonstrate the strong generalization of our method compared to VGCN and MC360IQA. Notably, during instances of training on the MVAQD dataset and subsequent testing on the IQA-ODI [46] dataset, our approach outperforms MC360IQA in terms of SRCC by an approximate margin of 0.8, and surpasses VGCN in SRCC by 0.5. Concurrently, when training on the CVIQD dataset, characterized by fewer distortion types, and testing on the MVAQD dataset, which encompasses more distortion types (JP2K, WN, GB), our approach attains elevated PLCC values of 0.5 and 0.3, respectively, as compared to MC360IQA and VGCN.

Table 5: Quantitative comparison of different generation methods (**RPS vs ScanGAN360 [24] and ScanDMM [32]**) with metrics of scanpath prediction task on JUFE [12] and JXUFE [33] datasets with authentic scanpaths.

| Published Time | Generation Method | JUFE [12] | | | JXUFE [33] | | |
|---|---|---|---|---|---|---|---|
| | | LEV↓ | DTW↓ | REC↑ | LEV↓ | DTW↓ | REC↑ |
| - | Random Baseline (lower bound) | 35.21 | 1707.45 | 0.38 | 35.08 | 1695.93 | 0.38 |
| TVCG22 | ScanGAN360 [24] | 32.53 | 1448.65 | 1.07 | 31.89 | 1427.55 | 1.14 |
| CVPR23 | ScanDMM [32] | 31.23 | **1434.36** | 1.21 | 31.48 | 1438.29 | 1.12 |
| - | RPS w/o DSP (Ours) | 29.54 | 1471.82 | 2.14 | 29.99 | 1463.38 | 1.94 |
| - | **RPS (Ours)** | **29.48** | 1454.03 | **2.21** | **29.66** | **1422.85** | **2.07** |
| - | Human Baseline (upper bound) | 23.85 | 1309.29 | 3.78 | 26.73 | 1302.15 | 2.88 |

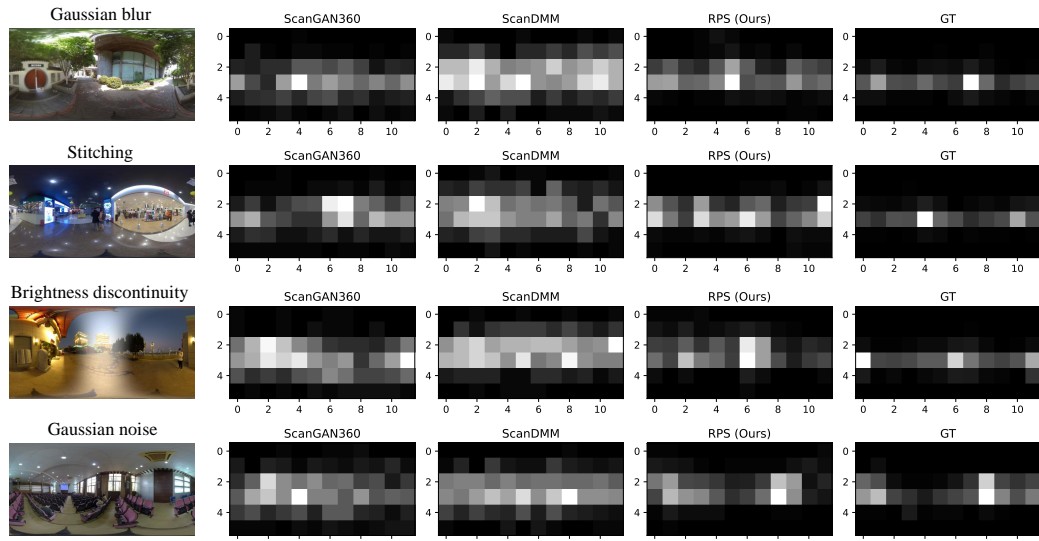

Figure 3: Visual comparison of the generated viewport positions for different methods on ODIs with four distortion types in JUFE. The brighter the area, the more viewports are generated in that area.

## 4.4 Effectiveness of Recursive Probability Sampling

As mentioned in Section 1, there exist many learning-based scanpath prediction methods [24, 47, 32]. They seem to be able to assist in constructing viewport sequences. However, they are hardly introduced to OIQA task. In this section, we first perform the quantitative comparison of the model with RPS and two advanced 360-degree scanpath prediction methods, namely ScanGAN360 [24] and ScanDMM [32] on the datasets without real observed scanpath. Subsequently, we compare the position and sequential order of viewports generated by the three methods with the ground-truth (GT) scanpaths by the metrics of scanpath prediction task and visualization to further validate the superiority of RPS on JUFE [12] and JXUFE [33] datasets with real observed scanpath.

**Quantitative comparison of performance.** We replace the proposed RPS method with Scan-GAN360 and ScanDMM to generate sequences of viewports. The model was trained and tested using these viewport sequences on OIQA [11] and MVAQD [18] datasets, maintaining the same length and number of viewports for a fair comparison. Table 4 shows the quantitative results, demonstrating that viewports generated from RPS yield superior performance compared to ScanGAN360 and ScanDMM. Additionally, training using viewports generated from these methods outperforms those generated using random schemes, highlighting the crucial role of suitable viewport sequences in the OIQA task.

Moreover, we conduct experiments by replacing original VGCN [43] sampling methods with RPS in VGCN. We use RPS to sample the same number of viewports as [43] to train VGCN on IQA-ODI [46] and MVAQD datasets. The results presented in Table 7 demonstrate a substantial performance enhancement for VGCN achieved by the viewport sampled through RPS, resulting in an increase of 0.07 in SRCC for MVAQD. Additionally, this indicates that the viewport sampled with RPS closely aligns with human observations.

Table 6: Quantitative comparison of different starting point positions on MVAQD [18] dataset.

| Position (latitude, longitude) | SRCC | PLCC |
|---|---|---|
| $(0°, 0°), (0°, 0°), (0°, 0°)$ | **0.9607** | **0.9720** |
| $(60°, 0°), (60°, 0°), (60°, 0°)$ | 0.9106 | 0.9312 |
| $(0°, 120°), (0°, 0°), (0°, -60°)$ | 0.9599 | 0.9660 |
| $(60°, 120°), (60°, 0°), (60°, -60°)$ | 0.9174 | 0.9455 |

Table 7: Quantitative comparison of using original VGCN sampling method and proposed RPS on IQA-ODI [46] and MVAQD [18] datasets.

| Method | IQA-ODI | | MVAQD | |
|---|---|---|---|---|
| | SRCC | PLCC | SRCC | PLCC |
| VGCN | 0.8117 | 0.8823 | 0.8422 | 0.9112 |
| VGCN-RPS | **0.8382** | **0.8883** | **0.9122** | **0.9273** |

Table 8: Ablation studies of each component in proposed Assessor360 on MVAQD [18] dataset.

| Method | Para (M) | SRCC | PLCC |
|---|---|---|---|
| Assessor360 w/o MFA | 88.53 | 0.8514 | 0.9171 |
| Assessor360 w/o DAB | 88.86 | 0.8437 | 0.8779 |
| Assessor360 w/ GAP | 88.69 | 0.9393 | 0.9587 |
| Assessor360 w/ GRU | 89.28 | **0.9607** | **0.9720** |

Table 9: Quantitative comparison of using GT sequences and sequences generated by RPS on JUFE [12] dataset. Starting Point (SP).

| Viewport Sequence | Good SP | | Bad SP | |
|---|---|---|---|---|
| | SRCC | PLCC | SRCC | PLCC |
| RPS (Ours) | 0.6623 | 0.6365 | 0.5044 | 0.4946 |
| GT Sequence | **0.7158** | **0.7013** | **0.5400** | **0.5377** |

**Comparison of viewport sampling positions and order.** Since each ODI in JUFE and JXUFE comprises 30 and 22 scanpaths, respectively, we employ the three methods to generate 30 and 22 sequences, each comprising 20 viewports, for each ODI in JUFE and JXUFE, respectively. For each GT sequence, we sample 20 viewports from 300 viewports with equal time intervals. We adopt Levenshtein distance (LEV) and dynamic time warping (DTW) metrics, as used in ScanGAN360 and ScanDMM, to evaluate the position and sequential order of the viewports. Furthermore, the viewing behaviors and patterns can be evaluated using the recurrence measure (REC) [24]. Each metric is calculated by comparing each generated sequence against all the GT sequences and computing the average, resulting in the final value. Besides, to establish an upper bound for each metric, we compute the Human Baseline by averaging the results of comparing each GT sequence against all other GT sequences [41]. Similarly, we establish a lower bound by randomly sampling viewports from the ODI, referred to as the Random Baseline.

The quantitative comparison results are reported in Table 5. The proposed unlearnable method outperforms learning-based models regarding the LEV and REC metrics. Specifically, it achieves a LEV improvement of 1.75 and 1.82 compared to ScanDMM [32], and a REC improvement of 1.14 and 0.93 compared to ScanGAN360 [24]. Additionally, our method achieves comparable performance to ScanDMM and ScanGAN360 on the DTW metric for both datasets. Figure 3 presents the positions of generated viewports for different generation methods. The panorama is divided into 72 regions, with each region spanning $15°$ degrees of latitude and $30°$ degrees of longitude. It can be observed that the positions generated by the learning-based method are relatively dispersed, while the positions generated by RPS are concentrated near the equator. The concentration gradually decreases with increasing latitude, aligning closely with the ground truth (GT) position.

### 4.5 Ablation Studies

**Impact of the initialization of the starting point.** For the ODI without real scanpath data, we initialize the coordinates of $N$ starting points to be $(0°, 0°)$ followed by [33]. We conduct experiments to assess the impact of different initial starting points positions on the final performance. Results shown in Table 6 revels that the model's performance displays relatively minor fluctuations during longitudinal shifts, yet experiences a substantial decline when changing larger latitudes (from points less frequented by humans as starting positions). This divergence arises due to the model's misalignment with authentic browsing behaviors (humans tend to start their panoramic observations closer to the equator). It further emphasizes that simulating actual scanpaths assists in more accurate image quality evaluation.

**Effectiveness of MFA and TMM.** We exclude the multi-scale feature aggregation (MFA) and distortion-aware Block (DAB) to assess their effectiveness in MVAQD [18]. Specifically, for the previous experiment, we utilize the final stage of the Swin Transformer [23] for feature output. The results, shown in Table 8, indicate that MFA and DAB substantially contribute to the performance, with only a minor increase in the network parameters. The GRU [5] module in TMM is replaced with global average pooling (GAP) for predicting the sequence quality score. The effectiveness of the GRU module in representing temporal transition information is demonstrated in Table 8, and it can assist the model in predicting more accurate quality scores.

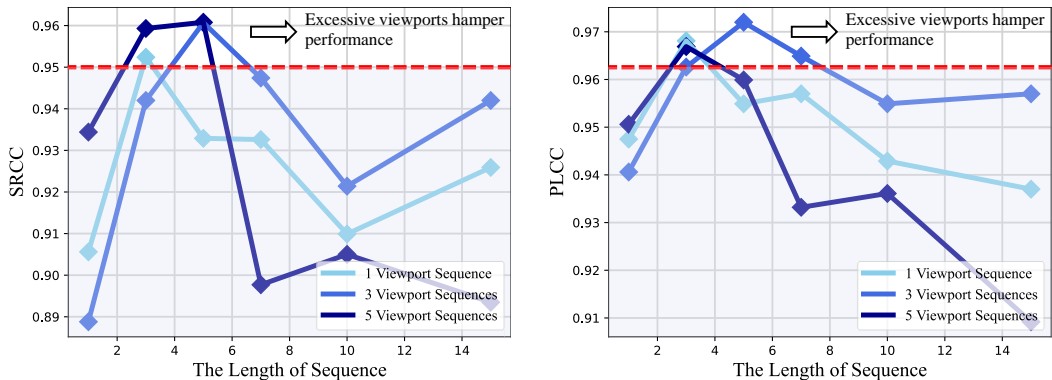

Figure 4: Performance of different number and length of viewport sequence on MVAQD [18].

**Impact of the number and length of the viewport sequence.** We test $N = 1, 3, 5$ three different numbers of viewport sequences with varying sequence lengths on MVAQD [18] dataset. The findings, shown in Figure 4, reveal that as the sequence length increases, there is a decreasing trend in model performance across all three numbers of sequences. This suggests that an excessive number of viewports may introduce redundant information, potentially disrupting the training process of the network. Furthermore, our experiments demonstrate that incorporating multiple viewport sequences, the model can capture a broader range of perspectives of the scene, thereby better reflecting the rating process of ODIs and achieving improved robustness.

### 4.6 Proximate to Human-Observed Performance

We conduct a comparative analysis between the performance achieved using ground-truth (GT) sequences and pseudo sequences generated by RPS on the JUFE dataset [12]. In the JUFE dataset, the GT sequences are annotated based on whether they originate from good or bad starting points (details in Supplementary Materials). Therefore, for each starting point, we use RPS to generate those sequences with the same number and length of sequences compared to GT sequences. Then, we apply the generated sequences and GT sequences as the input of our proposed network. The results shown in Table 9 highlight a close gap between the contributions of GT sequences and our generated sequences. This result emphasizes the significance of proximity to human observation in enhancing the model's capabilities. Meanwhile, there is still a large exploration space for future methods to better incorporate human's sequences in OIQA task.

## 5 Conclusion

This paper introduces a novel multi-sequence network named Assessor360 for BOIQA based on a realistic assessment procedure. Specifically, we design Recursive Probability Sampling (RPS) to generate viewport sequences based on the semantic scene and the distortion. Additionally, we propose Multi-scale Feature Aggregation (MFA) with Distortion-aware Block (DAB) to combine distorted and semantic features of viewports. Temporal Modeling Module (TMM) is introduced to learn the temporal transition of viewports. We demonstrate the high performance of Assessor360 on multiple OIQA datasets and validate the effectiveness of RPS by comparing it with two advanced learning-based models used for scanpath prediction. Limitation is that the transition direction and distance in RPS are fixed, resulting in equally spaced distances between viewports. However, we have confidence that our analyses and the proposed pipeline can provide long-term valuable insights for future OIQA task.

**Acknowledgments.** This work was partly supported by the National Natural Science Foundation of China (Grant No. 61991451) and the Shenzhen Science and Technology Program (JSGG20220831093004008). The author would like to thank Xiangjie Sui at Jiangxi University of Finance and Economics for his inspiration.

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
