# Supplementary Materials for Assessor360:
# Multi-sequence Network for Blind Omnidirectional Image Quality Assessment

**Tianhe Wu[1*], Shuwei Shi[1,2,*], Haoming Cai[3], Mingdeng Cao[2],**
**Jing Xiao[4], Yinqiang Zheng[2], Yujiu Yang[1†]**
[1] Shenzhen International Graduate School, Tsinghua University
[2] The University of Tokyo   [3] University of Maryland, College Park   [4] Pingan Group
{wth22, ssw20}@mails.tsinghua.edu.cn, cmd@g.ecc.u-tokyo.ac.jp
hmcai@umd.edu, xiaojing661@pingan.com.cn, yqzheng@ai.u-tokyo.ac.jp
yang.yujiu@sz.tsinghua.edu.cn

## A   Details of OIQA Datasets

The details of multiple datasets for OIQA task are presented in Table A. These datasets include CVIQD [23], OIQA [6], IQA-ODI [28], MVAQD [11], JUFE [8], and JXUFE [22]. Notably, CVIQD, OIQA, IQA-ODI, and MVAQD do not provide real scanpath data, whereas JUFE and JXUFE datasets contain actual user scanpath coordinates. For the dataset that contains scanpath coordinates, we can directly sample viewport sequences from it and use our network to predict the quality scores. However, it is challenging and costly to record user scanpath data for every ODI in realistic scenarios. The scanpath information is likely unavailable when evaluating the quality of a panorama. Therefore, we propose a generalized Recursive Probability Sampling (RPS) method to generate multiple pseudo viewport sequences for the panorama, which assists the network to predict an accurate quality score in a way that is similar to the observer's actual scoring process.

In JUFE and JXUFE, each ODI consists of 300 viewport coordinates, recorded using a head-mounted display (HMD). Besides, in both datasets, two different starting point coordinates are defined: a "good" starting point, where evaluators begin observing the ODI from a high-quality region, and a "bad" starting point, where the observing starting point is distorted. Additionally, evaluators provide two quality scores at 5 seconds and 15 seconds respectively, resulting in four Mean Opinion Score (MOS) labels for each ODI: 5s-bad, 5s-good, 15s-bad, and 15s-good.

## B   Calculation Process of the Scanpath Prediction Metric

We compare our proposed Recursive Probability Sampling (RPS) method with ScanGAN360 [16] and ScanDMM [21] using three metrics (details in Section 4.4): Levenshtein distance (LEV), dynamic time warping (DTW) and recurrence measure (REC) metrics as suggested in [7, 16, 21]. The higher REC values and lower LEV/DTW levels indicate greater prediction performance. The calculation process involves comparing each generated pseudo viewport sequence $s_i$ with each ground-truth (GT) sequence $\tilde{s}_j$ and averaging results. The calculation function can be formulated as:

$$\text{val} = \frac{1}{N_P} \frac{1}{N_G} \sum_{i=1}^{N_P} \sum_{j=1}^{N_G} f(s_i, \tilde{s}_j) \tag{1}$$

Here, $N_P$ and $N_G$ represent the number of pseudo and GT viewport sequences, respectively, and $f$ represents the function for calculating LEV, DTW, and REC. In our experimental setup, we ensure

---

[*]Tianhe Wu and Shuwei Shi contribute equally to this work.

[†]Corresponding author.

37th Conference on Neural Information Processing Systems (NeurIPS 2023).

Table A: The details of multiple OIQA datasets (CVIQD [23], OIQA [6], IQA-ODI [28], MVAQD [11], JUFE [8], JXUFE [22]).

| Name | Num. Ref/Dist Images | Resolution | Distortion Types | User Scanpath |
|---|---|---|---|---|
| CVIQD [23] | 16/528 | 4096×2048 | JPEG, H.264/AVC, H.265/HEVC | Unavailable |
| OIQA [6] | 16/320 | 11332×5666 to 13320×6660 | JPEG, JP2K, GB, GN | Unavailable |
| IQA-ODI [28] | 120/960 | 7680×3840 | JPEG, Projection | Unavailable |
| MVAQD [11] | 15/300 | 4K, 5K, 6K, 7K, 8K, 10K, 12K | JPEG, JP2K, HEVC, WN, GB | Unavailable |
| JUFE [8] | 258/1032 | 8192×4096 | GB, GN, BD, ST (Stitching) | Available |
| JXUFE [22] | 36/36 | 7680×3840 | ST, H.265 compression | Available |

Table B: The parameter information of learnable scanpath prediction methods (Param Ratio: scanpath prediction method parameter / Assessor360 parameter).

| Method | Params (M) | Param Ratio |
|---|---|---|
| DeepGaze III [13] | 78.9 | 88% |
| SaltiNet [1] | 103.6 | 116% |
| ScanGAN360 [16] | 33.9 | 38% |
| ScanDMM [21] | 18.7 | 21% |

Table C: Quantitative comparison of using different time series modeling modules in TMM on MVAQD [11] and OIQA [6] datasets.

| Method | MVAQD | | OIQA | |
|---|---|---|---|---|
| | SRCC | PLCC | SRCC | PLCC |
| Temporal Pooling [19] | 0.8942 | 0.9410 | 0.9566 | 0.9579 |
| LSTM [9] | 0.9261 | 0.9544 | 0.9701 | 0.9649 |
| Transformer [5] | 0.9368 | 0.9613 | 0.9761 | 0.9711 |
| GRU [2] | **0.9607** | **0.9720** | **0.9802** | **0.9747** |

that $N_P$ is equal to $N_G$. Specifically, the Human Baseline (upper bound is calculated by comparing each GT sequence $\tilde{s}_i$ against all the GT sequences).

## C   Functions of the Features at Different Stages of Swin Transformer

In our approach, we load the pre-trained weights of the Swin Transformer [15] on the ImageNet-1K dataset [3] and utilize the deep layers' features as abstract semantic information. When extracting features from deep networks, especially hierarchical architectures, the deep layers tend to capture more abstract representations, representing the overall semantic information of the image [15]. Indeed, the usage of deep features from hierarchical backbone as semantic information is a common practice in the field of image quality assessment [14, 31, 8, 33, 29, 20]. HyperIQA [20] also adopts a similar approach, utilizing deep features from hierarchical backbone to represent semantic information, while using shallow features to capture local distortion information. To further demonstrate this characteristic, we visualize the features at different stages of the Swin Transformer in Figure A. The results of the feature map clearly shows that the features of the first two stages primarily focus on local distortions. As the receptive field increases in the latter two stages, the feature map places more emphasis on the viewport semantic information.

## D   Discussion

### D.1   Difficulties in Applying Learnable Scanpath Prediction Methods

Deep learning-based scanpath prediction methods have not been extensively used in the OIQA task for several reasons: 1) Limited dataset size: OIQA datasets typically have a small amount of data, making it challenging to use deep neural networks to simulate observers' browsing process. This can lead to overfitting due to the large parameters of the pipeline. 2) Lack of scanpath labels: Many commonly used OIQA task datasets lack scanpath labels, making it difficult to perform supervised training for scanpath prediction models. 3) Difficult to optimize: The Table B presented illustrates that existing optimizable scanpath prediction models have a substantial quantity of parameters. This constitutes a minimum of 21% of the parameters within our OIQA model. Moreover, the current OIQA dataset contains a notably limited amount of data, posing challenges to optimizing the model with an increased parameter load, particularly when constrained by limited hardware resources.

### D.2   Advantages of RPS

We propose a novel sampling method called Recursive Probability Sampling (RPS), which is based on people's prior knowledge of the observation of scene semantics and local distortion characteristics. In comparison to previous sampling strategies used for the OIQA task [35, 8, 24, 27, 22, 31], our RPS method offers three distinct advantages: 1) Our proposed method generates multiple viewport sequences based on the characteristics of panoramas, which is not available in previous work. 2) Our method allows for sampling viewports from any starting point in a panorama, providing greater flexibility in real-world scenarios as shown in Figure C (a). 3) Even when the sampling process

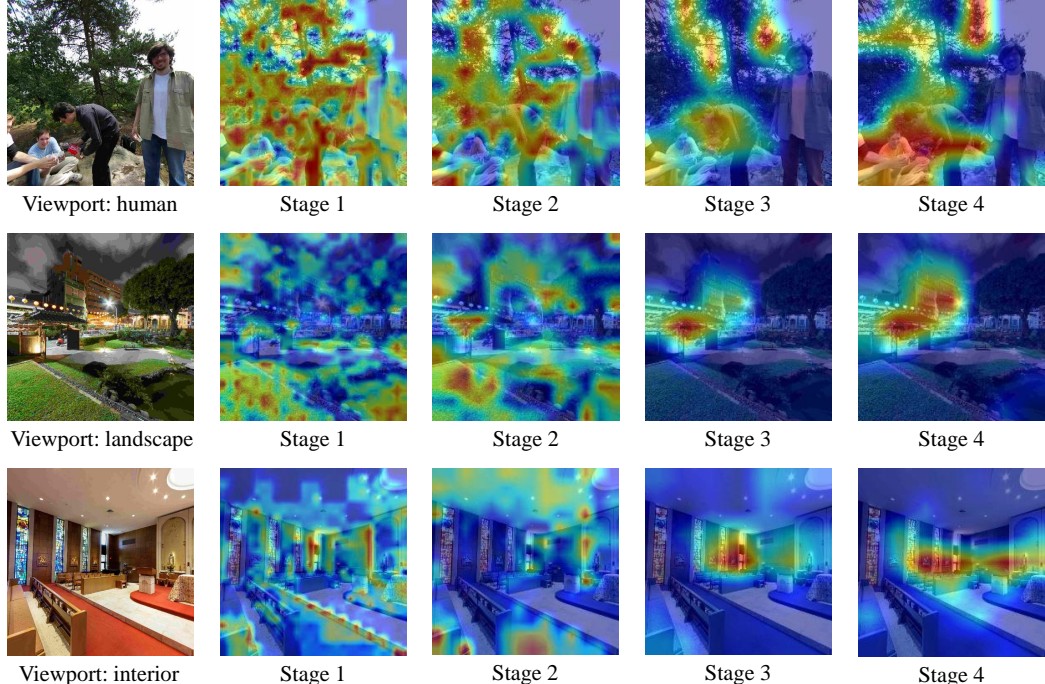

| Viewport: human | Stage 1 | Stage 2 | Stage 3 | Stage 4 |
| Viewport: landscape | Stage 1 | Stage 2 | Stage 3 | Stage 4 |
| Viewport: interior | Stage 1 | Stage 2 | Stage 3 | Stage 4 |

Figure A: Four-stage feature maps generated by Swin Transformer. We visualize the three most prevalent viewport contents in the omnidirectional image: human, landscape, and interior. The results of the feature map clearly shows that the features of the first two stages primarily focus on local distortions. As the receptive field increases in the latter two stages, the feature map places more emphasis on the semantic information within the viewport.

starts from the same starting point, it can produce different sampling routes through our method, as shown in Figure C (b). Our proposed RPS stimulate the real multi-assessor scoring process, enabling the model to learn the panorama's quality from diverse perspectives and enhance the accuracy and robustness of the model's evaluation capability.

### D.3 Limitation of RPS

We design a generalized RPS to generate multiple pseudo viewport sequences for ODIs, and numerous experiments have confirmed its effectiveness for the OIQA task. But, our current design has a limitation: the transition direction and distance are fixed, resulting in equally spaced distances between viewports. However, we still believe that our RPS can provide valuable insights for developing an appropriate method for generating viewport sequences to stimulate the action of the observers' browsing process in BOIQA task. In our future work, we plan to design a more effective generation method that can combine both details and content information to predict the flexible transition direction and distance, thereby addressing this limitation.

### D.4 More Observations

We draw several insightful observations from Table 1 in the main paper. 1) Previous learning-based NR methods, such as PaQ-2-PiQ [30], MUSIQ [12], MANIQA [29], CLIP-IQA [26], and LIQE [34] designed for 2D images, fail to handle panorama scenes due to the high-resolution characteristics of ODI. In contrast, traditional NR methods such as NIQE [18] and BRISQUE [17] demonstrate robust performance across all datasets, irrespective of resolution. Therefore, there is a need for novel approaches to effectively address the challenges posed by high-resolution ODI data in panorama scenes. 2) Several FR and NR methods, including WS-PSNR [25], WS-SSIM [36], VIF [32], MP-BOIQA [10], and MC360IQA [24], have shown effectiveness in addressing homogeneous distortions like compression present in CVIQD [23]. However, their assessment capability for multiple type distortions (JPEG, JP2K, GB, and GN) found in the OIQA dataset [6] is relatively weaker. Our proposed method is specifically designed for panorama scenes and demonstrates excellent evaluation capabilities for complex scenarios. This is achieved by ensuring that our pipeline aligns with the realistic assessment procedure.

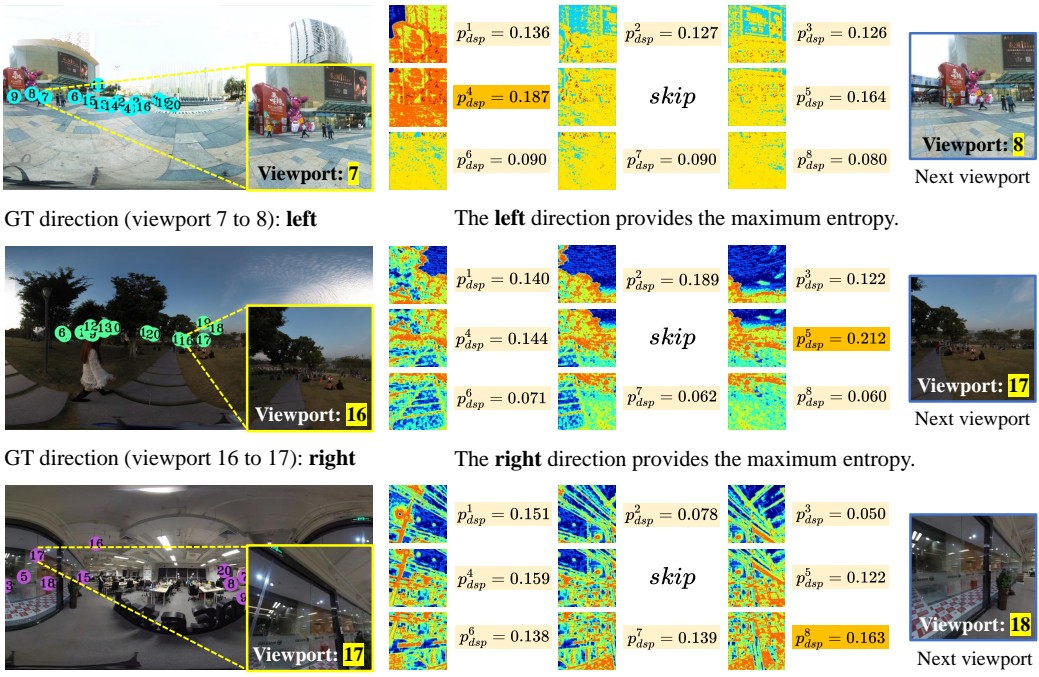

GT direction (viewport 7 to 8): **left**     The **left** direction provides the maximum entropy.

GT direction (viewport 16 to 17): **right**     The **right** direction provides the maximum entropy.

GT direction (viewport 17 to 18): **bottom right**     The **bottom right** direction provides the maximum entropy.

Figure B: The visualization of pixel-level entropy map. The visualization proves that entropy can accurately capture regions with high details. The sampling direction guided by entropy aligns consistently with the actual observational direction. It is important to note that in this context, entropy serves as the basis for sampling probability. The maximum entropy value is not adopted as the sampling direction.

## E  More Experimental Results

### E.1  Effectiveness of DSP

In image processing, information entropy is a metric used to measure the texture complexity of an image [4]. Therefore, our motivation is based on the observation that regions with denser texture information in an image tend to have higher information entropy, making them more attractive to human observers. In our DSP strategy, we assign higher probabilities to the corresponding transformation directions in order to emphasize these visually appealing areas. According to the findings presented in Table 5 of the main paper, DSP demonstrates a positive impact on a generation. Particularly, it significantly improves the DTW score by 17.79 and 40.53 on JUFE [8] and JXUFE [22], respectively.

We also visualize the entropy map of pixels to verify the motivation of this details-oriented sampling in Figure B. The visualization proves that entropy can accurately capture regions with high details and the sampling direction guided by entropy aligns consistently with the actual observational direction.

### E.2  Time Series Modeling Modules in TMM

We replace the GRU [2] of our method with Transformer [5], LSTM [9] and Temporal Pooling [19] to conduct the experiment on MVAQD [11] and OIQA [6] datasets. The results are displayed in Table C, highlighting GRU as the optimal performer. Our selection of GRU is justified by the following reasons: 1) In contrast to Temporal Pooling, the adaptive GRU exhibits superior temporal representation capabilities. 2) In comparison with LSTM and Transformer, GRU boasts a reduced parameter count, facilitating efficient training with limited data and shorter sequences while mitigating the risk of overfitting.

### E.3  More Visualization

We present additional quantitative comparison results of our Assessor360 with other state-of-the-art methods on two datasets, CVIQD [23] and IQA-ODI [28], to demonstrate its strong fitting ability.

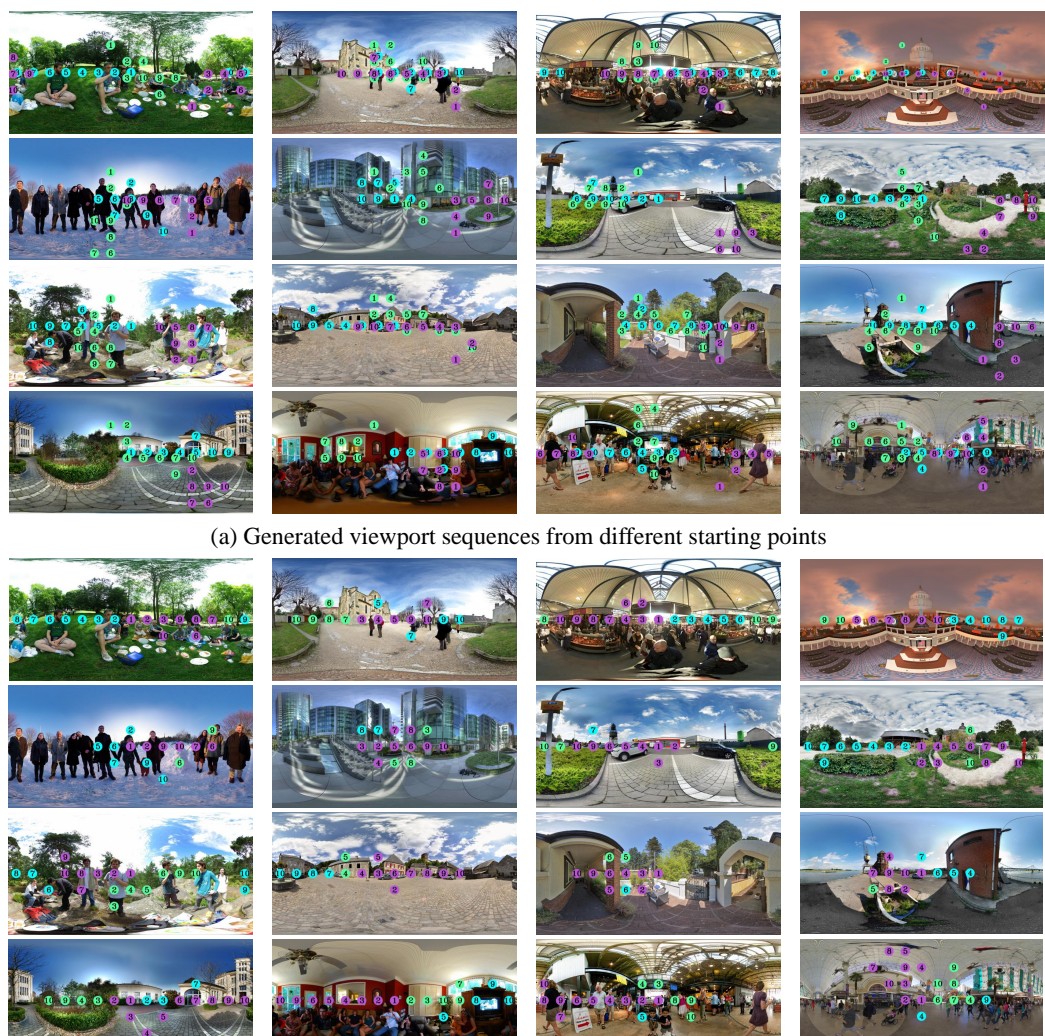

(a) Generated viewport sequences from different starting points

(b) Generated viewport sequences from the same starting point

Figure C: The visualization of generated viewport sequences by RPS. (a) We use RPS to generate three viewport sequences from different starting points. (b) We use RPS to generate three viewport sequences from the same starting point.

The comparison results are depicted in Figure D and Figure E. Existing methods show weaker fitting ability on the IQA-ODI test set than on the CVIQD test set. We attribute this to the higher content richness and resolution of the IQA-ODI dataset, which makes resolution-sensitive methods such as PSNR, WS-PSNR [25], WS-SSIM [36], etc., perform poorly. In contrast, our method simulates the real scoring process and constructs multiple viewport sequences, which allows the network to model the quality scores under different perspectives and thus give more consistent results. Additionally, methods such as MC360IQA [24], which do not consider the human scoring process, also demonstrate weaker performance.

Furthermore, we provide additional visualizations of the viewport sequences generated by our proposed RPS method, ScanGAN360 [16], and ScanDMM [21] to highlight the effectiveness of the RPS approach. The visualizations are depicted in Figure F. Our analysis reveals that GT viewports are mostly concentrated near the equator. Compared to ScanGAN360 and ScanDMM, our RPS method can focus on regions near the equator due to the added restriction of Equator-guided Sampled Probability (ESP). Additionally, our method can also focus on the distortion region, which aligns with GT viewports, owing to the introduction of Details-guided Sampled Probability (DPS), as shown in Figure F (c) bright regions. Conversely, the other two methods that are designed to focus on semantic information generate relatively dispersed regions, while neglecting regions with apparent distortion

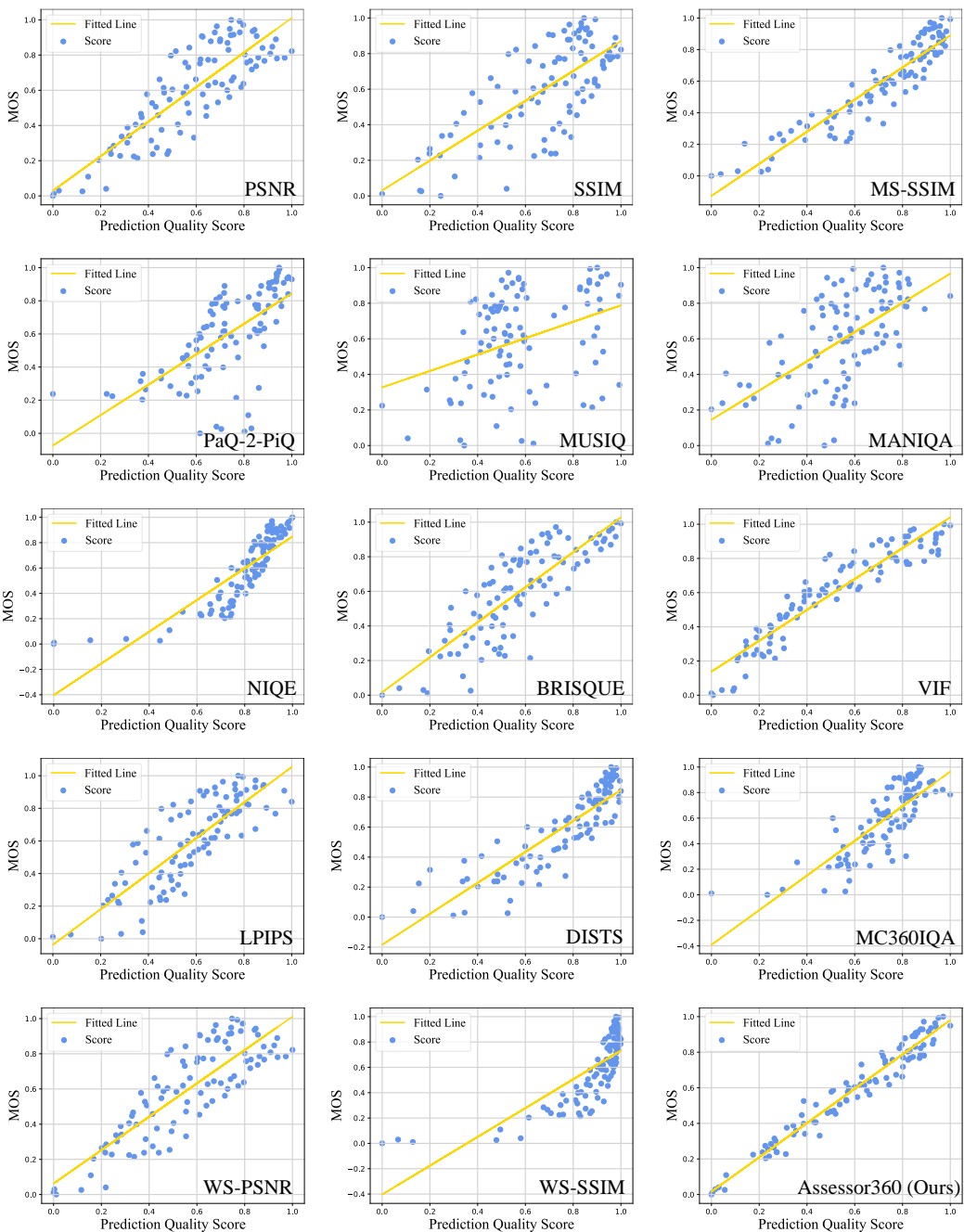

Figure D: The scatter plots of different SOTA methods' prediction quality scores versus the MOS values on CVIQD [23] testing set.

and rich details. Thus, our proposed RPS outperforms the existing methods by considering the scene semantics and local distortion characteristics of the ODI.

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

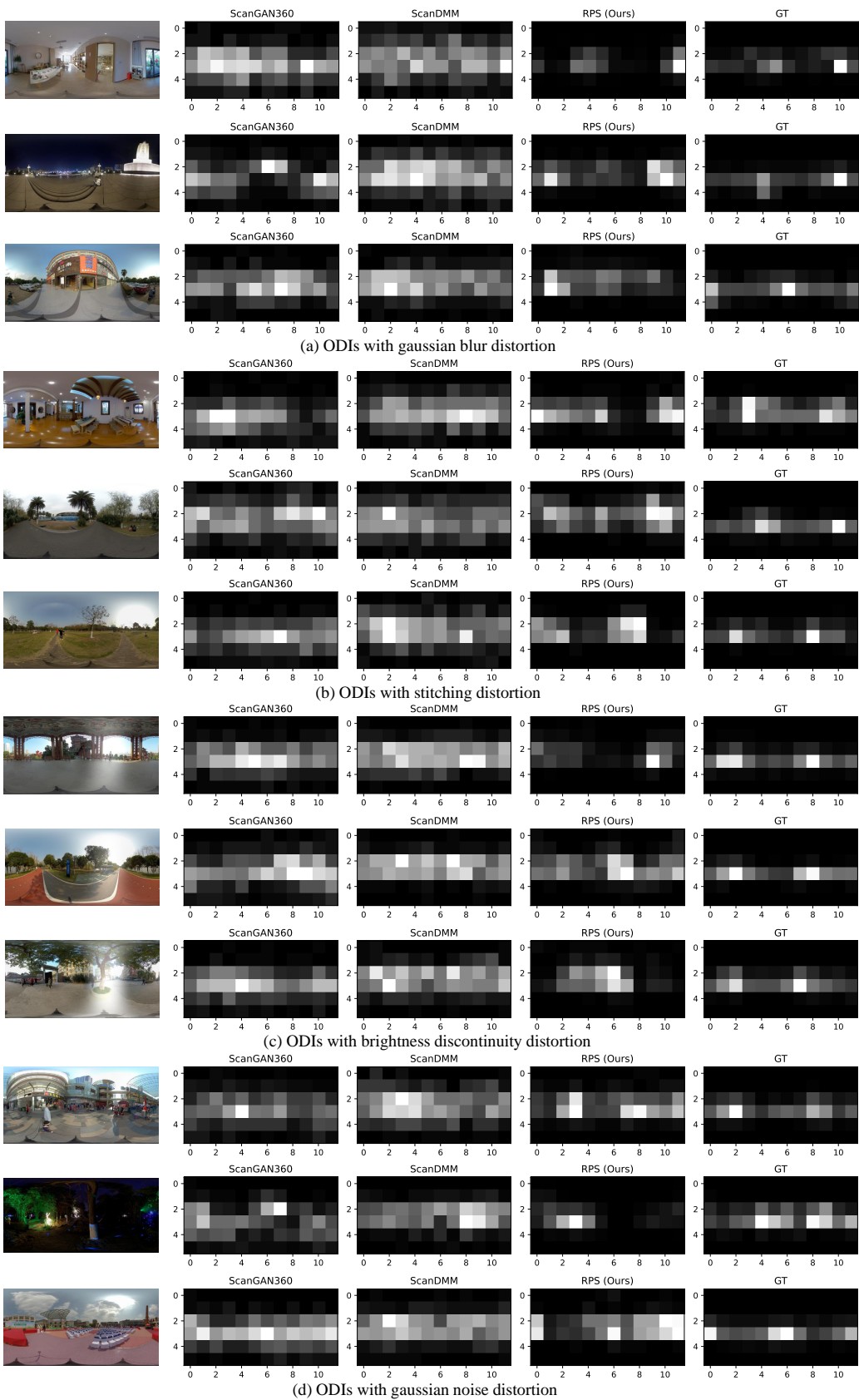

Figure F: More visual comparison of the generated viewport positions for different methods.