# OpenReview forum: "Assessor360: Multi-sequence Network for Blind Omnidirectional Image Quality Assessment"
_NeurIPS.cc/2023/Conference — NeurIPS 2023 poster_

### Official Review · Reviewer_BfkF · 2023-06-12

**Soundness:** 3 good
**Presentation:** 3 good
**Contribution:** 3 good
**Rating:** 7
**Confidence:** 5

**Summary:**

This paper introduces a BOIQA model called Assessor360, which takes into consideration the observer's browsing behavior. Specifically, RPS method is proposed to combine content and detailed information to generate multiple pseudo-viewport sequences from a given starting point. It also uses an MFA module with DAB to fuse distorted and semantic features. Finally, the viewport transition in the temporal domain is learned using TMM.

**Strengths:**

+ The pseudo multi-seqeuence generation is an interesting idea to mimic human behaviors during quality assessment.
+ The combination of MFA and DAB is reasonable in theory, and effective in practice.
+ Rich experiments are conducted on multiple datasets.

**Weaknesses:**

- To implement the TMM module, alternative design choices may be explored, such as LSTM, Transformer decoder, etc.
- L180-L182, the claims that the first two stages are more sensitive to distortion patterns and the latter capture abstract features are not verified.

**Questions:**

It would be interesting to compare the proposed method with the latest language-guided IQA models CLIPIQA[2] and LIQE[3].

[1] Wang, J., Chan, K. C., & Loy, C. C. (2022). Exploring CLIP for Assessing the Look and Feel of Images. AAAI 2023.

[2] Zhang, W., Zhai, G., Wei, Y., Yang, X., & Ma, K. (2023). Blind image quality assessment via vision-language correspondence: A multitask learning perspective. In Proceedings of the IEEE/CVF Conference on Computer Vision and Pattern Recognition (pp. 14071-14081).

**Limitations:**

Not applicable.

---

> ### Author Rebuttal · Authors · 2023-08-10
>
> We thank Reviewer #4 for his/her comments and the appreciation of our work. To the concerns expressed by Reviewer #4 in `Weaknesses` and `Questions`, our response is as follows:
>
> **`R4-Weakness 1`**: Thanks for your suggestion. We replace the GRU of our method with Transformer, LSTM and Temporal Pooling to conduct the experiment in the dataset. The results are displayed in the table below, highlighting GRU as the optimal performer. Our selection of GRU is justified by the following reasons: (1) In contrast to Temporal Pooling, the adaptive GRU exhibits superior temporal representation capabilities. (2) In comparison with LSTM and Transformer, GRU boasts a reduced parameter count, facilitating efficient training with limited data and shorter sequences while mitigating the risk of overfitting.
> |Method|MVAQD||OIQA||
> |-|-|-|-|-|
> ||SRCC|PLCC|SRCC|PLCC|
> |w/ Temporal Pooling|0.8942|0.9410|0.9566|0.9579|
> |w/ LSTM|0.9261|0.9544|0.9701|0.9649|
> |w/ Transformer|0.9368|0.9613|0.9761|0.9711|
> |w/ GRU|0.9607|0.9720|0.9802|0.9747|
>
> **`R4-Weakness 2`**: Thanks for your comments. In our approach, we load the pre-trained weights of the Swin Transformer on the ImageNet-1K dataset and utilize the shallow layers' features as local distortion information and deep layers' features as abstract semantic information. When extracting features from deep networks, especially hierarchical architectures, the deep layers tend to capture more abstract representations, representing the overall semantic information of the image [24]. Indeed, the usage of deep features from hierarchical backbone as semantic information is a common practice in the field of Image Quality Assessment [22, 50, 12, 53, 45, R1]. HyperIQA [R1] also adopts a similar approach, utilizing deep features from hierarchical backbone to represent semantic information, while using shallow features to capture local distortion information. To further demonstrate this characteristic, we visualize the features at different stages of the Swin Transformer in `attached file Figure B`. The results of the feature map clearly shows that the features of the first two stages primarily focus on local distortions. As the receptive field increases in the latter two stages, the feature map places more emphasis on the viewport semantic information.
>
> Additional References:
>
> [R1] Su, Shaolin, et al. "Blindly assess image quality in the wild guided by a self-adaptive hyper network." Proceedings of the IEEE/CVF Conference on Computer Vision and Pattern Recognition. 2020.
>
> **`R4-Q1`**: Thanks for your suggestion. The results of comparing our methods with ClipIQA and LIQE are shown in the tabel below.
> |Method|CVIQD||OIQA||IQA-ODI||MVAQD||
> |-|-|-|-|-|-|-|-|-|
> ||SRCC|PLCC|SRCC|PLCC|SRCC|PLCC|SRCC|PLCC|
> |ClipIQA+|0.4148|0.3333|0.1202|0.0933|0.0597|0.0695|0.3504|0.2621|
> |ClipIQA (resnet50)|0.4884|0.4347|0.2330|0.2531|0.0927|0.1929|0.5862|0.4941|
> |LIQE|0.8594|0.8086|0.7634|0.7419|0.8551|0.9020|0.6837|0.7539|
> |Assessor360|0.9644|0.9769|0.9802|0.9747|0.9573|0.9626|0.9607|0.9720|
>
> From the results in the above table, it can be seen that the performance of LIQE on the four OIQA datasets is much higher than that of ClipIQA. However, they calculate directly on the ERP format ODI, which exhibits geometric deformation at different latitudes. It will create a gap between the predicted scores and the GT scores. Our approach incorporates the ODI browsing process and possesses a robust capacity to depict distortion, thereby enabling it to attain exceptional performance in both fitting and generalization capabilities.

---

> > ### Comment · Reviewer_BfkF · 2023-08-11
> > **Post rebuttal**
> >
> > Additional results are provided, I am wiiling to improve my rating and recommend the authors to incorporate the additiona results into the final version.

---

### Official Review · Reviewer_rvUB · 2023-07-02

**Soundness:** 3 good
**Presentation:** 3 good
**Contribution:** 3 good
**Rating:** 6
**Confidence:** 3

**Summary:**

This manuscript proposes a novel multi-sequence network for BOIQA, which is inspired by the actual multi-assessor omnidirectional image quality assessment and introduces a generalized recursive probability sampling method for BOIQA tasks. In addition, a multi-scale feature aggregation module with distortion-aware blocks is designed. Extensive experimental results demonstrate that the proposed method achieves good performance on multiple OIQA datasets. However, there are still some problems in this manuscript.

**Strengths:**

With the development of VR-related techniques, there is an increasing need for omnidirectional image quality evaluation. However, the existing methods lack the modeling of the observer’s browsing process, causing the predicted score to be far from the human perceptual quality score. This paper proposes the first pipeline to simulate the authentic data scoring process in ODI quality assessment, and has certain innovation. Meanwhile, the logic of this manuscript is rigorous.

**Weaknesses:**

In my opinion, the most contribution of this paper is the recursive probability sampling strategy, while other modules are common. And some details are missing as follows:
1.	The input of the proposed model is in the form of sequences, but the Swin Transformer is for processing single images, hence how does the Transformer process multiple images in the sequence?
2.	In the recursive probability sampling process, how is the starting point of each sequence selected?
3.	The ODI’s quality is averaged by all predicted scores of each sequence, have you considered using weights to fuse these scores? How is the result?
4.	There are few comparative methods and experiments for cross-dataset testing.


**Questions:**

1.	The proposed model uses a pre-trained Swin Transformer, but the paper does not specify which kind of training model it is, whether it is tiny, small, or base.
2.	In the training phase, is the learning rate always kept at 1e-5?
3.	How are images tested during the testing phase? While choosing different starting point of each sequence, will the result be different? If the result will be different, how to obtain the final score?
4.	There is a typo, in lines 214 and 215, “SRCC and PLCC stand for Spearman rank-ordered correlation and Pearson linear correlation.” “coefficient” is missing. And the English usage and technical expression should be further polished.
5.	It is suggested that the numbers of the figures and tables in the paper be sorted according to the order of introduction.


**Limitations:**

The proposed RPS has a limitation: the transition direction and distance are fixed, resulting in equally spaced distances between viewports. Design a more effective generation method is author’s future work.

---

> ### Author Rebuttal · Authors · 2023-08-10
>
> We thank Reviewer #3 for his/her comments and the appreciation of our work. To the concerns expressed by Reviewer #3 in `Weaknesses` and `Questions`, our response is as follows:
>
> **`R3-Weakness 1`**: Thanks for your question. We utilize the Swin Transformer to process each viewport within the sampled sequence. Rather than processing them simultaneously, we independently extract features for each viewport and subsequently pass them into the Temporal Modeling Module (TMM) for temporal relationship modeling.
>
> **`R3-Weakness 2`**: Thanks for your question. For the ODI without real scanpath data, we initialize the coordinates of N starting points to be (0, 0) followed by [32]. These statements can be found in L203--L204 of the main text. In our experiments we found that increasing the number of starting points might potentially lead to better performance (see Figure 5), but it would also introduce additional computational overhead. We carefully considered the trade-off between the quantity of input viewports and performance and settled on selecting N=3. (More details can be seen in **`R2-Q1`**)
>
> **`R3-Weakness 3`**: Thanks for your question. We adopt a direct averaging approach based on the annotation process of the existing OIQA dataset [11,34]. During the data annotation process, multiple annotators provide ratings for each panoramic image, and the final mean opinion score (MOS) of a panoramic image is obtained by directly averaging the scores from all the annotators. Alternatively, if our focus shifts towards solely fitting the image quality score based on the model, we can conduct corresponding experiments involving the utilization of weighted fusion to amalgamate these scores. The experimental results presented in the table below reveal that there is not much difference between the two regression methodologies. The average approach is more stable when the number of scores is small (we only use three starting points).
> |Method|CVIQD||OIQA||IQA-ODI||MVAQD||
> |-|-|-|-|-|-|-|-|-|
> ||SRCC|PLCC|SRCC|PLCC|SRCC|PLCC|SRCC|PLCC|
> |Score-weight fusion|0.9815|0.9821|0.9651|0.9668|0.9528|0.9620|0.9183|0.9621|
> |Score-average|0.9644|0.9769|0.9802|0.9747|0.9573|0.9626|0.9607|0.9720|
>
> **`R3-Weakness 4`**: Thanks for your question. We conduct more cross-dataset testing to demonstrate the generalization capability of our method. We retrained MC360IQA, VGCN and our method, using 80% of the dataset for training and the remaining 20% for testing. We then select the model weights with the highest performance on the testset for cross-dataset testing for fair generalization comparisons.
>
> The SRCC results are shown in the following table:
> |Method|CVIQD|||OIQA|||MVAQD|||
> |-|-|-|-|-|-|-|-|-|-|
> ||OIQA|IQA-ODI|MVAQD|CVIQD|IQA-ODI|MVAQD|CVIQD|OIQA|IQA-ODI|
> |WS-PSNR|0.5027|0.4360|0.7225|0.7638|0.4360|0.7225|0.7638|0.5027|0.4360|
> |WS-SSIM|0.5442|0.5032|0.7930|0.6625|0.5032|0.7930|0.6625|0.5442|0.5032|
> |MC360IQA|0.4189|0.7114|0.0296|0.7044|0.5687|0.4081|0.0373|0.0025|0.0486|
> |VGCN|0.2361|0.2875|0.2452|0.6932|0.3873|0.4682|0.4650|0.6227|0.3921|
> |Assessor360|0.4597|0.8610|0.5640|0.8430|0.8751|0.6417|0.8756|0.7765|0.8646|
>
> The PLCC results are shown in the following table:
> |Method|CVIQD|||OIQA|||MVAQD|||
> |-|-|-|-|-|-|-|-|-|-|
> ||OIQA|IQA-ODI|MVAQD|CVIQD|IQA-ODI|MVAQD|CVIQD|OIQA|IQA-ODI|
> |WS-PSNR|0.4701|0.5468|0.6962|0.7895|0.5468|0.6962|0.7895|0.4701|0.5468|
> |WS-SSIM|0.4363|0.5941|0.6246|0.6536|0.5941|0.6246|0.6536|0.4363|0.5941|
> |MC360IQA|0.4295|0.7872|0.0404|0.7368|0.5930|0.4238|0.0430|0.0202|0.0646|
> |VGCN|0.2582|0.3127|0.2467|0.5929|0.3551|0.2419|0.3420|0.4642|0.3870|
> |Assessor360|0.5332|0.9032|0.5824|0.8636|0.9137|0.6565|0.7232|0.7287|0.8541|
>
> The results in the above table clearly demonstrate the strong generalization of our method compared to VGCN and MC360IQA. Notably, during instances of training on the MVAQD dataset and subsequent testing on the IQA-ODI dataset, our approach outperforms MC360IQA in terms of SRCC by an approximate margin of 0.8, and surpasses VGCN in SRCC by 0.5. Concurrently, when training on the CVIQD dataset, characterized by fewer distortion types, and testing on the MVAQD dataset, which encompasses more distortion types (JP2K, WN, GB), our approach attains elevated PLCC values of 0.5 and 0.3, respectively, as compared to MC360IQA and VGCN.
>
> **`R3-Q1`**: Thanks for your question. We use the **base** version of the pre-trained Swin Transformer.
>
> **`R3-Q2`**: Thanks for your question. Yes, it is always kept at 1e-5. In contrast to the three-stage training complexity of VGCN, our approach is less intricate. Maintaining learning rate to 1e-5 training can achieve good performance and generalization.
>
> **`R3-Q3`**: In the testing phase, the model's hyperparameters and the quantity of viewports sampled using RPS are maintained at the same values as those employed during the training phase. Initially, employ RPS to generate N scanpaths from the coordinate (0, 0), subsequently forwarding these scanpaths to the network for inference.
>
> Choosing different starting points will generate different viewports, and the results will definitely be different (for detailed analysis, see **`R2-Q1`**). Indeed, some distortions (such as stitching) and non-uniform distortion can lead to substantial discrepancies in quality scores assigned by observers situated at different starting points [12, 32]. Therefore, **for the same image in the JUFE and JXUFE datasets, two MOS scores are calculated as labels for the two starting points**. If the necessity arises to restrict an image to a singular quality score, the viable approach is to compute the average quality score.
>
> **`R3-Q4`**: Thanks for your suggestion. We will make the revisions and improve the English usage and expression according to your suggestions.
>
> **`R3-Q5`**: Thanks for your suggestion. We will make the necessary revisions according to your feedback to make the paper appear more standardized.

---

> > ### Comment · Reviewer_rvUB · 2023-08-12
> >
> > Some questions have been solved and  additional results are provided, I will keep my rating.

---

### Official Review · Reviewer_hPRU · 2023-07-05

**Soundness:** 2 fair
**Presentation:** 2 fair
**Contribution:** 2 fair
**Rating:** 6
**Confidence:** 5

**Summary:**

A blind quality assessment model is proposed for 360-degree images.

**Strengths:**

1. The problem is interesting and practical.
2. The attempt to model the observer’s browsing process is a plus.
3. The results look promising.

**Weaknesses:**

1. The recursive probability sampling is a little ad hoc to the reviewer. It is mostly handcrafted, which is not optimized to mimic the observer's browsing process. This weakens the motivation of the proposed method.
2. No validation set is used for the hyperparameter setting, making the results look like overfitting (surpassing full-reference models by a large margin).
3. Since no validate set is available, more cross-dataset results are necessary to justify the rationality of the proposed model design.
4. Two baselines with ERP and cube map projection as direct input should be compared.

**Questions:**

1. How to initialize the starting points (the total number and positions), and does the initialization affect the final performance?
2. How to interpret Fig. 4?

**Limitations:**

N.A.

---

> ### Author Rebuttal · Authors · 2023-08-09
>
> We thank Reviewer #2 for his/her comments. Our responses are as follows:
>
> **`R2-Weakness 1`**: Thanks for your comment. We respectfully **DISAGREE** with your opinion that the handcrafted RPS weakens our motivation.
>
> Firstly, it is true that previous OIQA methods have relied on handcrafted viewport sampling from ODIs [40,50], followed by subsequent processing. Our approach leverages the content distribution of panoramic images and incorporates human visual fixation characteristics to simulate the human browsing process, making it more flexible and reasonable. **Handcrafted methods for viewport sampling from panoramic images have indeed become a common practice in the field and do not weaken our motivation.**
>
> Secondly, deep learning-based scanpath prediction methods have not been extensively used in the OIQA task for several reasons: **(1) Limited dataset size**: OIQA datasets typically have a small amount of data, making it challenging to use deep neural networks to simulate observers' browsing process. This can lead to overfitting due to the large parameters of the pipeline. **(2) Lack of scanpath labels**: Many commonly used OIQA datasets lack scanpath labels, making it difficult to perform supervised training for scanpath prediction models. **(3) Difficult to optimize**: The table presented below illustrates that existing optimizable scanpath prediction models have a substantial quantity of parameters. This constitutes a minimum of 21% of the parameters within our OIQA model. Moreover, the current OIQA dataset contains a notably limited amount of data, posing challenges to optimizing the model with an increased parameter load, particularly when constrained by limited hardware resources.
> |Method|Params (M)|Param ratio: Sampling Method / Assessor360|
> |-|-|-|
> |DeepGaze III|78.9|88%|
> |SaltiNet|103.6|116%|
> |ScanGAN|33.9|38%|
> |ScanDMM|18.7|21%|
> |RPS|0.0|0.0%|
>
> Finally, our method exhibits strong generalization and universality, as evident from the additional table on generalization experiments. Compared to other handcrafted sampling methods, RPS achieves superior performance, and simply replacing the sampling method in VGCN with RPS leads to significant improvements (see **`R1-Q3`**). This substantiates the effectiveness of our RPS strategy.
>
> **`R2-Weakness 2 and R2-Weakness 3`**: Thanks. In addition to the cross-dataset experiment, we retrained MC360IQA, VGCN, and our method, using 80% of the dataset for training and the remaining 20% for testing. We then select the model weights with the highest performance on the test set for cross-dataset testing for fair generalization comparisons. The SRCC values are shown in the table below (see **`R3-Weakness 4`** for more results).
> |Method|CVIQD|||OIQA|||MVAQD|||
> |-|-|-|-|-|-|-|-|-|-|
> ||OIQA|IQA-ODI|MVAQD|CVIQD|IQA-ODI|MVAQD|CVIQD|OIQA|IQA-ODI|
> |MC360IQA|0.4189|0.7114|0.0296|0.7044|0.5687|0.4081|0.0373|0.0025|0.0486|
> |VGCN|0.2361|0.2875|0.2452|0.6932|0.3873|0.4682|0.4650|0.6227|0.3921|
> |Assessor360|0.4597|0.8610|0.5640|0.8430|0.8751|0.6417|0.8756|0.7765|0.8646|
>
> These experimental results illustrate the outstanding generalization performance of our model, indicating that it is not just outperforming the FR model due to overfitting. In fact, there are hardly any deep learning-based FR-OIQA quality assessment methods. As a result, the FR methods listed in our paper are primarily proposed for 2D-IQA scenarios. They calculate directly on the ERP format ODI, which exhibits geometric deformation at different latitudes. It will create a gap between the predicted scores and the GT scores. WS-PSNR and WS-SSIM face challenges in aligning with the HVS and exhibit limited capability in characterizing image distortion features. Our approach incorporates the ODI browsing process and possesses a robust capacity to depict distortion, thereby enabling it to attain exceptional performance in both fitting and generalization capabilities.
>
> **`R2-Weakness 4`**: Thanks for your suggestion.  We will involve these two baselines in the paper.
> |Method|CVIQD||OIQA||IQA-ODI||MVAQD||
> |-|-|-|-|-|-|-|-|-|
> ||SRCC|PLCC|SRCC|PLCC|SRCC|PLCC|SRCC|PLCC|
> |w/ ERP|0.9330|0.9485|0.8961|0.8857|0.9098|0.9196|0.9076|0.9240|
> |w/ Cube map|0.9390|0.9412|0.9216|0.9170|0.9105|0.9122|0.8966|0.9324|
>
> **`R2-Q1`**:  Different initializations (total number and positions) of the starting point can indeed impact the final performance. For the ODI without real scanpath data, we initialize the coordinates of N starting points to be (0, 0) followed by [32]. In our experiments, we found that increasing the number of starting points might potentially lead to better performance (see Figure 5), but it would also introduce additional computational overhead. We carefully considered the **trade-off** between the number of input viewports and performance and settled on selecting N=3. We also conducted tests to assess the impact of different initial positions on the final performance. Humans tend to start their panoramic observations closer to the equator. The model's performance displays relatively minor fluctuations during longitudinal shifts, yet experiences a substantial decline when changing larger latitudes (from points less frequented by humans as starting positions). This divergence arises due to the model's misalignment with authentic browsing behaviors.
> |Position (latitude, longitude)|SRCC|PLCC|
> |-|-|-|
> |(0, 0), (0, 0), (0, 0)|0.9607|0.9720|
> |(60, 0), (60, 0), (60, 0)|0.9106|0.9312|
> |(0, 120), (0, 0), (0, -60)|0.9599|0.9660|
> |(60, 120), (60, 0), (60, -60)|0.9174|0.9455|
>
> **`R2-Q2`**: We depict the spatial distribution of 15 GT scanpaths within the ODI, with each scanpath encompassing the coordinates of 20 visited viewports. The brighter the area in the figure, the higher the frequency of visits in this area. The purpose of visualizing this representation is to facilitate an intuitive comparison of the accuracy of sampling locations obtained through different sampling methods against the GT.

---

> > ### Comment · Reviewer_hPRU · 2023-08-12
> > **Final comment**
> >
> > Thank the authors for handling the reviewer's comments with a detailed and convincing rebuttal. Aftering taking other reviewers' comments (and the corresponding response) into account, the reviewer is willing to uplift the rating.

---

### Official Review · Reviewer_7uSg · 2023-07-06

**Soundness:** 2 fair
**Presentation:** 3 good
**Contribution:** 2 fair
**Rating:** 5
**Confidence:** 4

**Summary:**

This paper proposed a novel blind omnidirectional image quality assessment framework based on the modeling of the observer’s browsing process, which is called Assessor360. In detail, Assessor360 designs a novel viewpoint sampling methods named “Recursive probability sampling (RPS)” based on content-guided-sampled probability and detailed-guided sampled probability. In addition, the authors utilize the Multi-scale Feature Aggregation (MFA) to fuse the shallow layers’ feature of Swin transformer which is enhanced by proposed distortion-aware block (DAB) and deep layer’s feature which can capture abstract feature.

**Strengths:**

1. The motivation is clear and the performance is promising.

**Weaknesses:**

In general, novelty and insight of the paper is limited to some extent. Below are some questions:
1.   The main contribution of Assessor360 is the human browsing related viewpoint sampling methods, which combine CSP and DSP. Why is sampling via the equator bias named content-guided? Generally speaking, it may be easier to understand if object information or semantic information is used as content guidance. In DSP, the author did not visualize the entropy map of pixels to verify the motivation of this detail-oriented sampling
2.	In section3.3, the author did not explain how the semantic information was extracted? Or the deep layer’s feature of swin transformer can represent the semantic information? Without constraints on deep layer’s features or adding conditions, it is difficult to understand that deep features have extracted semantic information.
3.	RPS can be understood as a plug-and-play sampling method. Has the author tried other methods, such as VGCN, with the RPS sampling method to verify the effect?
4.	It would be better for the authors to try and analysis some other time series modeling modules in TMM, such as transformer, etc.


**Questions:**

Please refer above comments

---

> ### Author Rebuttal · Authors · 2023-08-09
>
> We thank Reviewer #1 for his/her insightful comments. Our responses are as follows:
>
> **`R1-Q1`:** Thanks for the questions. For your `first question`, we agree that the name "content-guided" may be commonly used when the method is guided by the object information and semantic information in the traditional 2D scenario. However, in the context of panoramic scene explored in our paper, **"content-guided" refers to the bias in information density across different latitudes of the panorama**. A common observation is that content information becomes denser as we move closer to the equatorial region of the panoramic image in most instances [9]. At the same time, It is also indicated in [30] that users tend to fixate around the equator of the panoramas, with very few fixations in latitudes far from it, which can influence the final quality assessment results. To model this prior knowledge, we use equator bias as a guidance prior to calculate the probabilities for scene transitions in the RPS strategy. Hence, we refer to it as Content-guided Sampled Probability (CSP). To further prove this phenomenon, we use the "Spatial Information (SI)" metric to conduct the statistic experiment to show the information density distribution across different latitudes of the panorama. The data presented in the table supports the conclusion that the density of information within the panorama exhibits a gradual decline from the equator towards the poles.
> |Latitude|CVIQD|OIQA|IQA-ODI|MVAQD|
> |-|-|-|-|-|
> |54~90|387|2210|3174|814|
> |18~54|950|2386 |5177|1290|
> |-18~18|1560|2576|7019|1604|
> |-54~-18|1010|2463|6480|1496|
> |-90~-54|894|2177|5979|1246|
>
> For the `second question`, in image processing, information entropy is a metric used to measure the texture complexity of an image [7].  Therefore, our motivation is based on the observation that regions with denser texture information in an image tend to have higher information entropy, making them more attractive to human observers. In our DSP strategy, we assign higher probabilities to the corresponding transformation directions in order to emphasize these visually appealing areas. We visualize the entropy map of pixels to verify the motivation of this detail-oriented sampling in the `attached file Figure A`. The visualization proves that entropy can accurately capture regions with high detail and the sampling direction guided by entropy aligns consistently with the actual observational direction.
>
> **`R1-Q2`:** Thanks for the question. In our approach, we load the pre-trained weights of the Swin Transformer on the ImageNet-1K dataset and utilize the deep layers' features as abstract semantic information. When extracting features from deep networks, especially hierarchical architectures, the deep layers tend to capture more abstract representations, representing the overall semantic information of the image [24]. **Indeed, the usage of deep features from hierarchical backbone as semantic information is a common practice in the field of Image Quality Assessment [22, 50, 12, 53, 45, R1].** HyperIQA [R1] also adopts a similar approach, utilizing deep features from hierarchical backbone to represent semantic information, while using shallow features to capture local distortion information. To further demonstrate this characteristic, we visualize the features at different stages of the Swin Transformer in `attached file Figure B`. The results of the feature map clearly shows that the features of the first two stages primarily focus on local distortions. As the receptive field increases in the latter two stages, the feature map places more emphasis on the viewport semantic information.
>
> Additional References:
>
> [R1] Su, Shaolin, et al. "Blindly assess image quality in the wild guided by a self-adaptive hyper network." Proceedings of the IEEE/CVF Conference on Computer Vision and Pattern Recognition. 2020.
>
> **`R1-Q3`:** Thanks for the question. First, we do **NOT** argue that RPS can be understood as a plug-and-play sampling method in our paper. The key components of our method involve two main steps:  RPS performed in a specific order and then using the TMM to model the sequential relationships among the sampled viewports. **The combination of these two steps is essential for achieving the desired effect (see main paper Table 5 and L292--L295).** However, the possibility of achieving better performance with viewports obtained solely through RPS sampling, compared to previous sampling methods, remains a valid point. In order to delve deeper into this matter, we further conduct experiments by replacing their sampling methods with RPS in VGCN. We use RPS to sample the same number of viewports as [40] to train VGCN. The results presented in the table below demonstrate a substantial performance enhancement for VGCN achieved by the viewport sampled through RPS, resulting in an increase of +0.07 in SRCC for MVAQD. Additionally, this indicates that the viewport sampled using RPS closely aligns with human observations.
> |Method|IQA-ODI||MVAQD||
> |-|-|-|-|-|
> ||SRCC|PLCC|SRCC|PLCC|
> |VGCN|0.8117|0.8823|0.8422|0.9112|
> |VGCN-RPS|0.8382|0.8883|0.9122|0.9273|
>
> **`R1-Q4`:** Thanks for your suggestion. We replace the GRU of our method with Transformer, LSTM and Temporal Pooling to conduct the experiment in the dataset. The results are displayed in the table below, highlighting GRU as the optimal performer. Our selection of GRU is justified by the following reasons: **(1)** In contrast to Temporal Pooling, the adaptive GRU exhibits superior temporal representation capabilities. **(2)** In comparison with LSTM and Transformer, GRU boasts a reduced parameter count, facilitating efficient training with limited data and shorter sequences while mitigating the risk of overfitting.
> |Method|MVAQD||OIQA||
> |-|-|-|-|-|
> ||SRCC|PLCC|SRCC|PLCC|
> |w/ Temporal Pooling|0.8942|0.9410|0.9566|0.9579|
> |w/ LSTM|0.9261|0.9544|0.9701|0.9649|
> |w/ Transformer|0.9368|0.9613|0.9761|0.9711|
> |w/ GRU|0.9607|0.9720|0.9802|0.9747|

---

> > ### Comment · Reviewer_7uSg · 2023-08-13
> > **Final comment**
> >
> > Some of the concerns have been addressed.

---

> > > ### Author Response · Authors · 2023-08-14
> > > **Further discussion with Reviewer 7uSg (denoted as R1)**
> > >
> > > Dear Reviewer 7uSg:
> > >
> > > Thanks again for your precious time and valuable comments.
> > >
> > > Should you have any additional inquiries or suggestions regarding our paper, please do not hesitate to inform us. We are very happy to offer further explanations and engage in enriching discussions.
> > >
> > > We hold the strong belief that our contribution holds value for the community, a sentiment shared by the other three reviewers. Initially, Reviewer rvUB (denoted as R3) and Reviewer BfkF (denoted as R4) both thought positively about our work. After rebuttal, Reviewer hPRU (denoted as R2) now agrees that our response solves his/her concerns and rates our work acceptable.
> > >
> > > We found that you and other reviewers share some similar concerns. The concerns include several points as follows:
> > >
> > > (1) The proof of deep features representing semantic information. (See the details in the attached file)
> > >
> > > (2) Analysis of other time series modeling modules in TMM.
> > >
> > > (3) Discussion of the novelty and insight of our paper. (See the details in "Author Rebuttal by Authors")
> > >
> > > (4) Other details and concerns of our method.
> > >
> > > Thank you once more for your valuable time and insightful feedback.
> > >
> > > Thanks,
> > >
> > > Paper 4933 Authors.

---

### Author Rebuttal · Authors · 2023-08-10

Dear Reviewers and ACs:

We sincerely appreciate all the reviews. They give positive and high-quality comments on our paper with a lot of constructive feedback.
We would like to emphasize that our work proposes a novel, simple and effective new pipeline called Assessor360. We appreciate your recognition of our pioneering effort in attempting to simulate the real panorama quality assessment process, aligning it with the evaluation performed by actual human raters. Our novelty and contribution can be summarized as following:

(1) It is crucial to emphasize that our method can simulate the browsing process of human observers on OIQA datasets, especially on those lack real observed scanpaths through our carefully designed sampling strategy, and has achieved state-of-the-art performance. The paradigm proposed is indeed novel for the community. We first demonstrate the significance of simulating the real scoring process for the OIQA task through our proposed method. In fact, under certain distortions (stitching), observers will give very different quality scores according to their observing process. This is also the primary reason why certain OIQA methods that neglect the sequence of observations yield subpar performance on the JUFE dataset. Future OIQA research can benefit from our findings.

(2) We propose a novel and efficient sampling method called RPS, which sequentially samples salient viewports. It is the first time a viewport sampling method leverages both scene characteristics and local texture and distortion properties to mimic the observer's real evaluation process. Through our approach, we compare RPS with deep learning-based sampling methods. Our method outperforms the deep learning-based methods in terms of performance while requiring fewer parameters.

(3) It is worth noting that ours is the first method to address the OIQA task using a novel multi-sequence pipeline. In the actual data annotation process, we obtain the final label by averaging evaluation scores from multiple different observers who have observed the panorama. Unlike previous approaches that simply model a fixed observation order and directly regress them to the final label, our multi-sequence based method effectively captures this complex process.

We have supplemented our work with additional experiments and visual analyses to incorporate the insightful suggestions of the reviewers:

According to Reviewer 7uSg’s concerns, we visualize the entropy map of pixels to verify the motivation of our proposed detail-oriented sampling in Figure A of the attached file.

According to Reviewer 7uSg and Reviewer BfkF’s questions, we visualize the four-stage feature maps generated by Swin Transformer.

According to Reviewer 7uSg and Reviewer BfkF’s concerns, we add the analytical experiment on TMM with different modules in the rebuttal area.

According to Reviewer 7uSg's question, we add the experiments on the effects of RPS in VGCN.

According to Reviewer hPRU and Reviewer rvUB's concerns, we add more cross-dataset testing experiments in the rebuttal area.

According to Reviewer hPRU and Reviewer rvUB's questions, we have conducted a thorough analysis of the impact of starting points on the final performance and supplementary experiments to support our findings in the rebuttal area.

According to Reviewer hPRU and Reviewer BfkF's suggestions, we add more methods to compare with our results in the rebuttal area.

According to Reviewer rvUB's suggestions, we will make the necessary revisions according to your feedback to make the paper appear more standardized.

In the final version, we will improve other minor points of Reviewer 7uSg, Reviewer hPRU, Reviewer rvUB, and Reviewer BfkF. Thank you all for the valuable suggestions.

Thanks,

Paper 4933 Authors.

---

### Decision · Program_Chairs · 2023-09-21

**Decision:**

Accept (poster)

**Comment:**

Reviewers raised issues and offered suggestions regarding novelty, alternatives to sampling and time series modeling, generalization, and comparisons with baseline methods in the initial reviews. Rebuttals provide additional visualizations and experiments requested by the reviews. The reviewers generally agree that the rebuttal is convincing and are optimistic about the paper. Most reviews appreciate the idea of modeling observers' browsing behavior for quality assessment. Moreover, the RPS sampling method is novel, and the combination of MFA and DAB is effective. Experiments show that the proposed method surpasses the STOA performance and significantly outperforms the compared methods.